# A revised model of TRAIL-R2 DISC assembly explains how FLIP(L) can inhibit or promote apoptosis

Luke M Humphreys, Jennifer P Fox, Catherine A Higgins, Joanna Majkut, Tamas Sessler, Kirsty McLaughlin, Christopher McCann, Jamie Z Roberts, Nyree T Crawford, Simon S McDade [iD], Christopher J Scott, Timothy Harrison & Daniel B Longley* [iD]

## Abstract

The long FLIP splice form FLIP(L) can act as both an inhibitor and promoter of caspase-8 at death-inducing signalling complexes (DISCs) formed by death receptors such as TRAIL-R2 and related intracellular complexes such as the ripoptosome. Herein, we describe a revised DISC assembly model that explains how FLIP(L) can have these opposite effects by defining the stoichiometry (with respect to caspase-8) at which it converts from being anti- to pro-apoptotic at the DISC. We also show that in the complete absence of FLIP(L), procaspase-8 activation at the TRAIL-R2 DISC has significantly slower kinetics, although ultimately the extent of apoptosis is significantly greater. This revised model of DISC assembly also explains why FLIP's recruitment to the TRAIL-R2 DISC is impaired in the absence of caspase-8 despite showing that it can interact with the DISC adaptor protein FADD and why the short FLIP splice form FLIP(S) is the more potent inhibitor of DISC-mediated apoptosis.

**Keywords** apoptosis; caspase-8; DISC; FLIP; TRAIL-R2
**Subject Category** Autophagy & Cell Death

## Introduction

Apoptosis can be activated through the intrinsic (mitochondrial-mediated) and extrinsic (death receptor-mediated) pathways and is frequently dysfunctional in cancer [1,2]. The extrinsic apoptotic pathway is initiated when death ligands such as TRAIL (tumour necrosis factor (TNF)-related apoptosis-inducing ligand) bind their receptors (TRAIL-R1 (DR4)/TRAIL-R2 (DR5)). This binding enables the intracellular death domains (DDs) of the receptors to engage the adaptor protein FADD (Fas-associated protein with death domain) through homotypic interactions with its DD [3–7]. Engagement of the FADD DD by the activated receptor enables its other domain, the death effector domain (DED), to engage in homotypic protein–protein interactions with other DED-containing proteins, most

importantly the tandem DED-containing procaspase-8 (FADD-like interleukin-1β-converting enzyme, FLICE) and its regulator FLIP (FLICE-inhibitory protein) [8–10]. At this complex, known as the death-inducing signalling complex (DISC), procaspase-8 forms homodimers or heterodimers with FLIP via homotypic interactions between their DEDs.

Homo-dimerisation is required for activation and processing of procaspase-8, which occurs in a two-step manner: rate-limiting inter-dimer cleavage between adjacent dimers recruited to the DISC that cleaves the linker between its large (p18) and small (p10) catalytic domains, followed by intra-dimer cleavage between the p18-subunit and the DED-containing pro-domain. The 2nd cleavage step releases the active enzyme made up of a hetero-tetramer of two p18 and two p10 subunits from the DISC, which can then propagate an apoptotic signal by cleaving downstream substrates such as BID and procaspases-3/7 [11–16].

At the DISC, hetero-dimerisation of FLIP(S) with procaspase-8 prevents formation of the caspase-8 active site and thereby inhibits procaspase-8 processing [17]. However, FLIP(L) has a pseudo-catalytic domain, and when it hetero-dimerises with procaspase-8, this results in formation of an enzymatically active complex with altered substrate specificity compared to the caspase-8 homodimer [18], which is retained at the DISC [19]. The role of FLIP(L) at the DISC is complex, and whether it acts to promote or inhibit caspase-8 activation is a matter of debate. In this report, we determine the stoichiometry of FLIP(L):procaspase-8 at the DISC that results in apoptosis induction versus apoptosis inhibition and provide data that advance our understanding of the interplay between FLIP and caspase-8 in this complex and which supports a revised model of DISC assembly.

## Results

### Assessment of TRAIL-R2 DISC stoichiometry

In a previous study, we quantified the relative levels of FLIP, caspase-8 and FADD at the TRAIL-R2 DISC using recombinant protein standards [20]. The approximate ratio of 2 molecules of

The Patrick G. Johnston Centre for Cancer Research, Queen's University Belfast, Belfast, UK
*Corresponding author. Tel: +44 2890 972647; E-mail: d.longley@qub.ac.uk

every tandem DED protein (caspase-8 and FLIP) for every 1 molecule of FADD was in disagreement with those obtained by two other groups [21,22], who noted higher ratios of tandem DED proteins to FADD. To reconcile these differing findings, we developed a new quantification approach using Flag-tagged proteins of similar molecular weight to endogenous proteins (expressed and purified from human cells rather than truncated versions derived from bacterial expression systems) to quantitatively evaluate the stoichiometry of FLIP, caspase-8 and FADD at the TRAIL-R2 DISC (Fig EV1) [20]. In addition, we used both N- and C-terminal antibodies to detect fully processed caspase-8 (the DED1/2-only pro-domain) retained at the DISC, a form of caspase-8 not quantified in our earlier study.

Three cancer cell lines representing 3 cancer types, A549 (NSCLC), HCT116 (colon) and DU145 (prostate), were selected to assess TRAIL-R2 DISC stoichiometry. A549 is a relatively TRAIL-resistant NSCLC cell line [23], while HCT116 is a TRAIL-sensitive model of colorectal cancer, and DU145 is a classical model of androgen-independent prostate cancer representing a later stage of the disease and which typically expresses high levels of FLIP [24]. Basal protein expression (Fig EV2A) and sensitivity (caspase activation and cell death) to a recombinant multivalent form of recombinant TRAIL (IZ-TRAIL, which can activate both TRAIL-R1 and TRAIL-R2; Fig EV2B) confirmed that these models express the relevant TRAIL-R2 pathway components and that the HCT116 model is more sensitive than the A549 model, with the DU145 model in between, giving us a good range of models with which to study TRAIL-R2 DISC assembly.

To assess the stoichiometry of FLIP, caspase-8 and FADD, a TRAIL-R2 DISC IP was performed in cells treated with increasing concentrations (1×, 2×, 4×) of anti-TRAIL-R2 antibody (AMG655, conatumumab) conjugated to magnetic beads to enable efficient receptor cross-linking, thereby mimicking the receptor clustering effects of both the endogenous membrane-bound ligand and 2nd generation multivalent TRAIL-R2-targeting therapeutics [20,25–27]. In all 3 cell lines, the cleavage products of procaspase-8 (p41/43 and p24/26) increased from 1× to 2× AMG655 (Fig 1A and C). However, going from 2× to 4×, there was little change, suggesting saturation of binding to available (i.e. cell surface) TRAIL-R2 receptors; this was supported by analysis of TRAIL-R2 in the bound and unbound fractions (Fig 1A and B). A similar pattern of DISC recruitment was observed for FADD and both splice forms of FLIP (Figs 1A and C, and EV2E). Assessment of the unbound fraction revealed increasing levels of the p18-caspase-8 large catalytic subunit (indicative of full procaspase-8 processing to the active hetero-tetramer) from 1× to

2×, which correlated with cleavage of the caspase-3 target PARP (a marker of apoptosis induction; Fig 1B). In line with this, analysis of caspase-8-like (IETDase) activity in the unbound fraction indicated that increasing from 1× to 2× AMG655 led to an increase in activity in all 3 cell lines, but not between the 2× and 4× samples (Fig 1D). The caspase activity and PARP cleavage analyses in the unbound fractions also indicated that, as expected, the HCT116 and DU145 models were more sensitive to the AMG655 beads than the A549 model, and these results were also reflected in cellular caspase activity and apoptosis assays (Fig EV2C). Moreover, these results were recapitulated with the 2nd generation multivalent TRAIL-R2-specific agonist, MEDI3039 (Fig EV2D).

Quantitative comparison of the relative levels of procaspase-8 and FADD at the DISC indicated that the ratio remained relatively constant within each cell line, regardless of the level of TRAIL-R2 stimulation (Fig 1E). In the most TRAIL-sensitive HCT116 cells, the ratio of caspase-8 (the sum of the p55, p41/43 and p24/26 subunits) to FLIP (p43-FLIP(L) plus FLIP(S)) was highest (~5–6:1), while it was lowest in the more TRAIL-resistant A549 model (close to 1:1). When the ratios of the tandem DED proteins (caspase-8 plus FLIP) were quantified relative to FADD, they were remarkably consistent across all 3 models for all 3 levels of receptor stimulation, with the average ratio calculated to be ~3:1 (Fig 1E); thus, including analysis of the p24/26-caspase-8 pro-domains in our quantitative DISC analyses resulted in ratios of tandem DED proteins to FADD that are higher than those calculated in our previous study (~2:1) [20], although still less than some of the ratios reported by others (up to 9:1) [21,22,28]. Moreover, excluding the p24/26-caspase-8 pro-domain from these analyses yielded a ratio of tandem DED proteins to FADD of ~2.2:1, similar to our original study, indicating that both DISC quantification approaches (using recombinant proteins or Flag-tagged proteins as standards) are valid.

We next calculated how expression of DISC protein components correlated with caspase-8-like activity across all 3 cell line models. Of note, a highly significant correlation was observed between caspase-8 activity and the ratio of FADD:FLIP(L) (Pearson's $r = 0.91$; $P = 0.0006$; Fig 1F), while the correlation between caspase-8 activity and the ratio of total caspase-8:FLIP(L) approached significance (Pearson's $r = 0.67$; $P = .0506$). This suggests that the relative levels of FLIP(L) recruited to the DISC are the major determinant of the extent of TRAIL-R2-induced caspase-8 activity.

To determine whether caspase-8 processing altered DISC stoichiometry, we assessed the effect of the pan-caspase inhibitor

**Figure 1.  Assessment of TRAIL-R2 DISC stoichiometry.**

A   Western blot analysis of Caspase 8, FLIP and FADD recruitment to the TRAIL-R2 DISC in A549, HCT116 and DU145 cell lines following incubation with increasing concentrations (1×, 2×, or 4×) of anti-TRAIL-R2 (AMG655)-conjugated magnetic beads for 90 min.

B   Western blot analysis of the unbound soluble fraction from (A). An untreated input of each cell line is included for comparison.

C   Quantification of Western blot analysis from panel (A) for FLIP, FADD, procaspase-8 and their respective cleavage fragments at the TRAIL-R2 DISC. Proteins were quantified by densitometry and normalised to known protein standards (see Fig EV1).

D   Caspase-8-like (IETDase) activity assay in the unbound soluble fraction from the DISC IP in panel (A). An untreated control (0) was used for each cell line to show basal caspase-8 activity.

E   Ratios of DED proteins quantified at the TRAIL-R2 DISC from panel (C) in A549, HCT116 and DU145 cells; increasing concentrations of AMG655 are indicated by colour intensity.

F   Correlation between caspase-8 activity and ratio of FADD:FLIP(L).

Source data are available online for this figure.

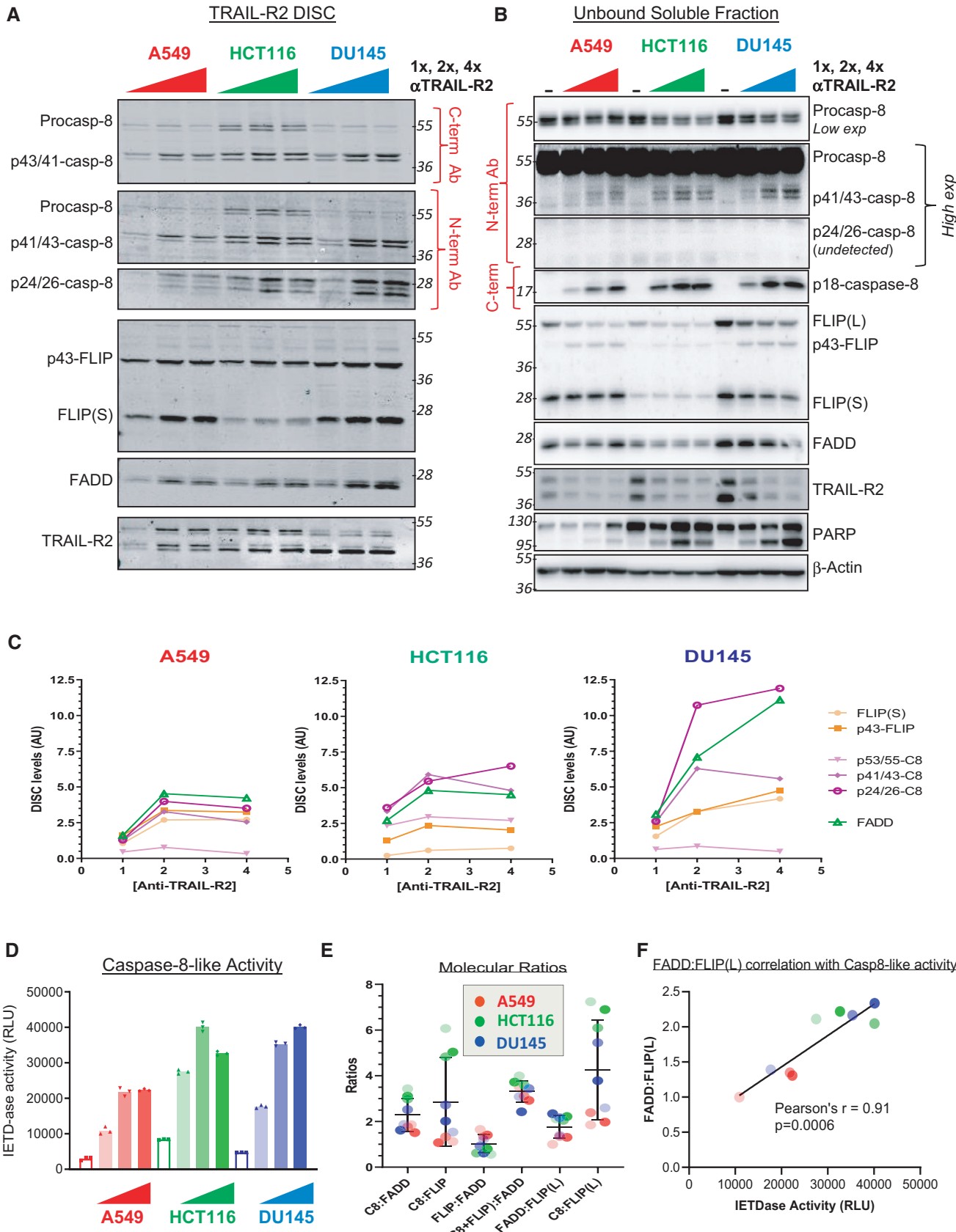

**Figure 1.**

zVAD-fmk (Fig EV3A–C). Despite partial inhibition of caspase-8 and FLIP(L) processing, the ratios between the 3 main DISC components were largely unaffected by zVAD-fmk, with the ratio of FADD to total caspase-8 plus total FLIP again averaging around 3 (Fig EV3G). This ratio was observed in several independent experiments, as was the tendency for caspase-8:FADD and caspase-8:FLIP ratios to be highest and FLIP:FADD ratios to be lowest in the TRAIL-sensitive HCT116 model (Fig EV3D–G).

Importantly, these results are not consistent with the caspase-8 chain model of DISC assembly [21,22,28], which proposes formation of caspase-8 DED1/2-dominated chains of variable lengths depending on the relative levels of each DISC component available and extent of receptor activation. Not only did we detect a lower ratio of the tandem DED proteins (caspase-8 plus FLIP) to FADD (meaning shorter "chains"), we also found that these ratios varied little across multiple models with varying levels of DISC component expression (Fig EV2A) and extent of receptor activation (average = 3.0; SD = 0.6; $n$ = 16; Fig EV3G).

### FLIP(L) inhibits full caspase-8 activation when equi-stoichiometric at the DISC

We next used siRNA targeting procaspase-8 to titrate the ratio of procaspase-8 to FLIP in the TRAIL-sensitive HCT116 model. Transfection of siCASP8 resulted in a concentration-dependent reduction in procaspase-8 and its cleavage products at the DISC (Figs 2A–C and EV4A). Surprisingly, as caspase-8 levels at the DISC decreased, no significant reductions in the recruitment of either splice form of FLIP were observed (Fig 2A and B). Quantification of caspase-8 and FLIP revealed a ~7:1 ratio in control (SC) cells in keeping with the results presented above for the HCT116 model (Figs 1E and EV3C and E). Notably, as the ratio of caspase-8 to FLIP approached 1:1, caspase-8 (IETDase) and downstream caspase-3/7 (DEVDase) activity approached the baseline levels in untreated cells (Fig 2D). To avoid potentially confounding effects from the intrinsic apoptotic pathway on DISC processing events and caspase activity measurements, we also used a HCT116 BAX/BAK knockout (KO) model (Fig EV4C–F); again, as the ratio of caspase-8:FLIP approached 1:1, caspase-8 and caspase-3/7 activity approached baseline levels. In addition, we found highly significant correlations between the levels of caspase-8 cleavage products (p41/43 and p24/26) at the DISC and IETDase activity in the unbound soluble fraction (Fig EV4B), indicating that this enzymatic activity reflects caspase-8

activity and providing additional validation of the quantification processes used.

As a complementary approach, we assessed the impact of increasing FLIP levels relative to caspase-8 at the DISC using a prostate cancer cell line (PC3) stably expressing exogenous FLIP (L). The ratio of caspase-8:FLIP at the DISC was high in the control EV cell line even compared to the HCT116 model (Fig 2E and F). In the FLIP(L) overexpressing model, this was reduced and correlated with a decrease in caspase-8 activity (Fig 2G). Upon treatment with the HDAC inhibitor MS275, endogenous FLIP proteins were down-regulated (consistent with previous studies [29]); however, the exogenous FLIP(L) (not under control of the endogenous promoter) was unchanged. Subsequently, the ratio of caspase-8:FLIP was significantly increased in the EV control model in response to MS275; although despite this increase, there was no change in caspase-8 activity (Fig 2G). In the FLIP(L) overexpressing model treated with MS275, the ratio of caspase-8 to FLIP (now almost entirely FLIP(L) as endogenous FLIP(S) was down-regulated) fell just below 1 and caspase-8 activity was significantly inhibited (Fig 2F). Collectively, these results indicate that when the levels of FLIP(L) and procaspase-8 recruited to the DISC are approximately equal, FLIP(L) acts as an inhibitor of caspase-8 activation.

### Although FLIP can interact directly with FADD in a caspase-8-independent manner, its DISC recruitment is highly caspase-8-dependent

In the siRNA experiments (Fig 2), while procaspase-8 was significantly down-regulated at the highest concentrations of siRNA used, it could still be detected at the DISC. To assess the impact of *complete* loss of procaspase-8 on DISC assembly, we next generated a number of CRISPR-Cas9 *CASP8* deletion models. In agreement with the findings of others for the TRAIL-R1 and Fas/CD95 DISCs reported during the completion of these studies [28,30], we found that *CASP8* deletion markedly inhibited recruitment of FLIP to the TRAIL-R2 DISC (Figs 3A and EV5A). However, at later timepoints (180 min), there were detectable, albeit low, levels of FLIP (L) at the DISC in *CASP8* null cells, approximately half of which was in its unprocessed p55-form (Fig 3A; *lane 6*). In agreement with the findings of others for the Fas/CD95 DISC [30], recruitment of procaspase-8's other paralog, procaspase-10, to the TRAIL-R2 DISC was also inhibited in the absence of procaspase-8;

---

**Figure 2. Impact of decreasing procaspase-8 on DISC assembly and caspase activation.**

A  Western blot analysis of FADD, FLIP and caspase-8 at the TRAIL-R2 DISC in HCT116 cells treated with escalating doses (0–30 nM) of caspase-8 siRNA for 48 h. Samples were incubated with anti-TRAIL-R2 (AMG655) beads (4×) for 90 min prior to collection. A scrambled control siRNA (SC) was transfected (30 nM) for comparison.

B  Quantification of caspase-8 (p55, p43/41 and p26/24) and FLIP (FLIP(L), FLIP(S) and p43-FLIP) at the TRAIL-R2 DISC by densitometry and normalised to known protein standards (see Fig EV1).

C  Ratio of caspase-8:FLIP calculated from values presented in (B).

D  Caspase-8 (IETDase) and caspase-3/7 (DEVDase) activity assay in the unbound soluble fraction from panel (A).

E  Western blot analysis of FLIP, caspase-8, and FADD recruitment to the TRAIL-R2 DISC in PC3 cells stably expressing an empty vector (EV) or Flag-tagged FLIP L (FL) treated with 2.5 μM MS-275 for 48 h, followed by a 90-min DISC IP. * modified form of FLIP, potentially mono-ubiquitinated p43-FLIP(L).

F  Table of the relative ratios of caspase-8 (p55, p41/43 and p24/26) and FLIP (endogenous and Flag-tagged) at the TRAIL-R2 DISC in MS-275-treated and untreated cells after a 90-min DISC IP.

G  Caspase-8 (IETDase) activity assays of the unbound soluble fraction from (E).

Source data are available online for this figure.

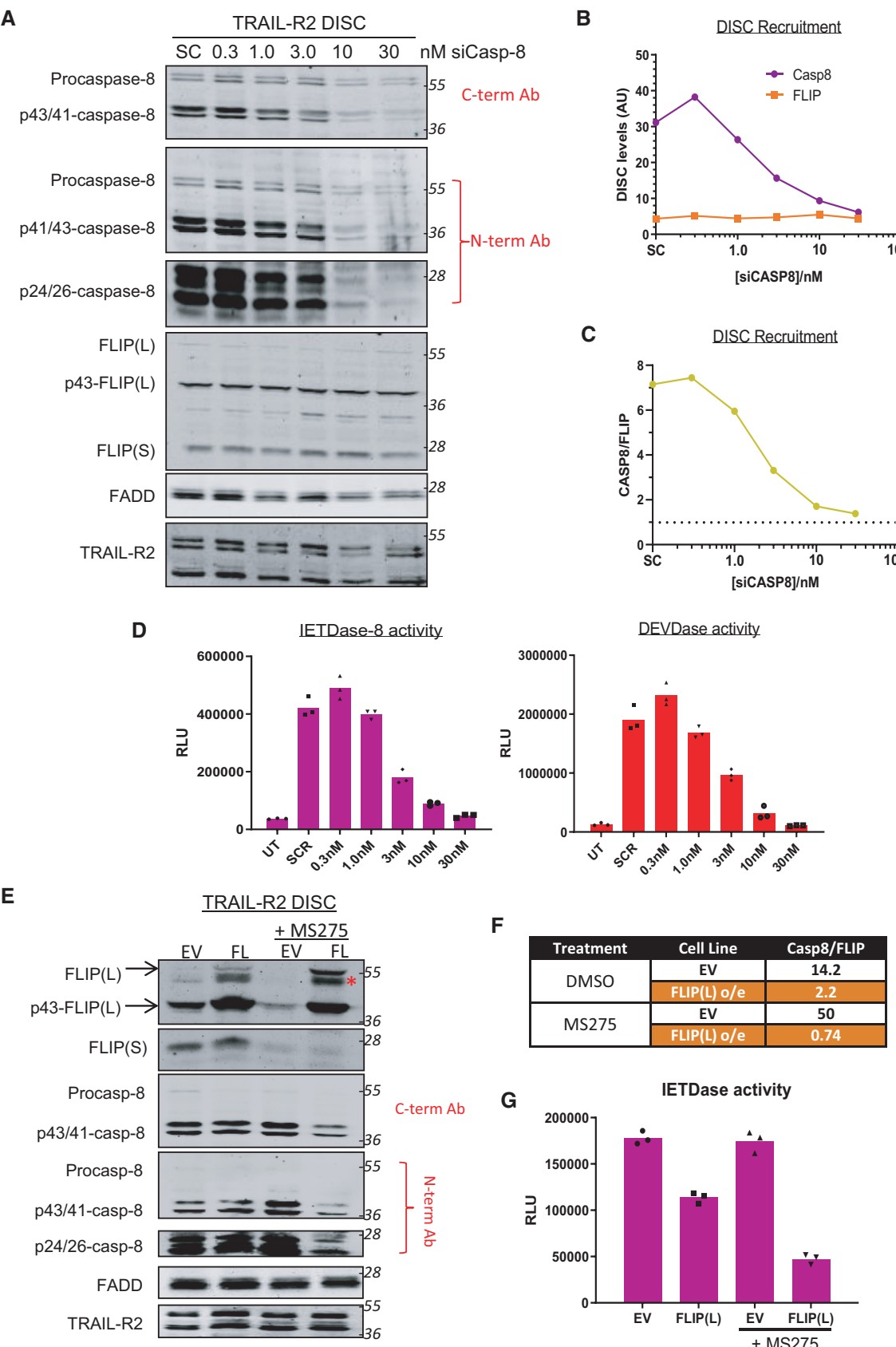

**Figure 2.**

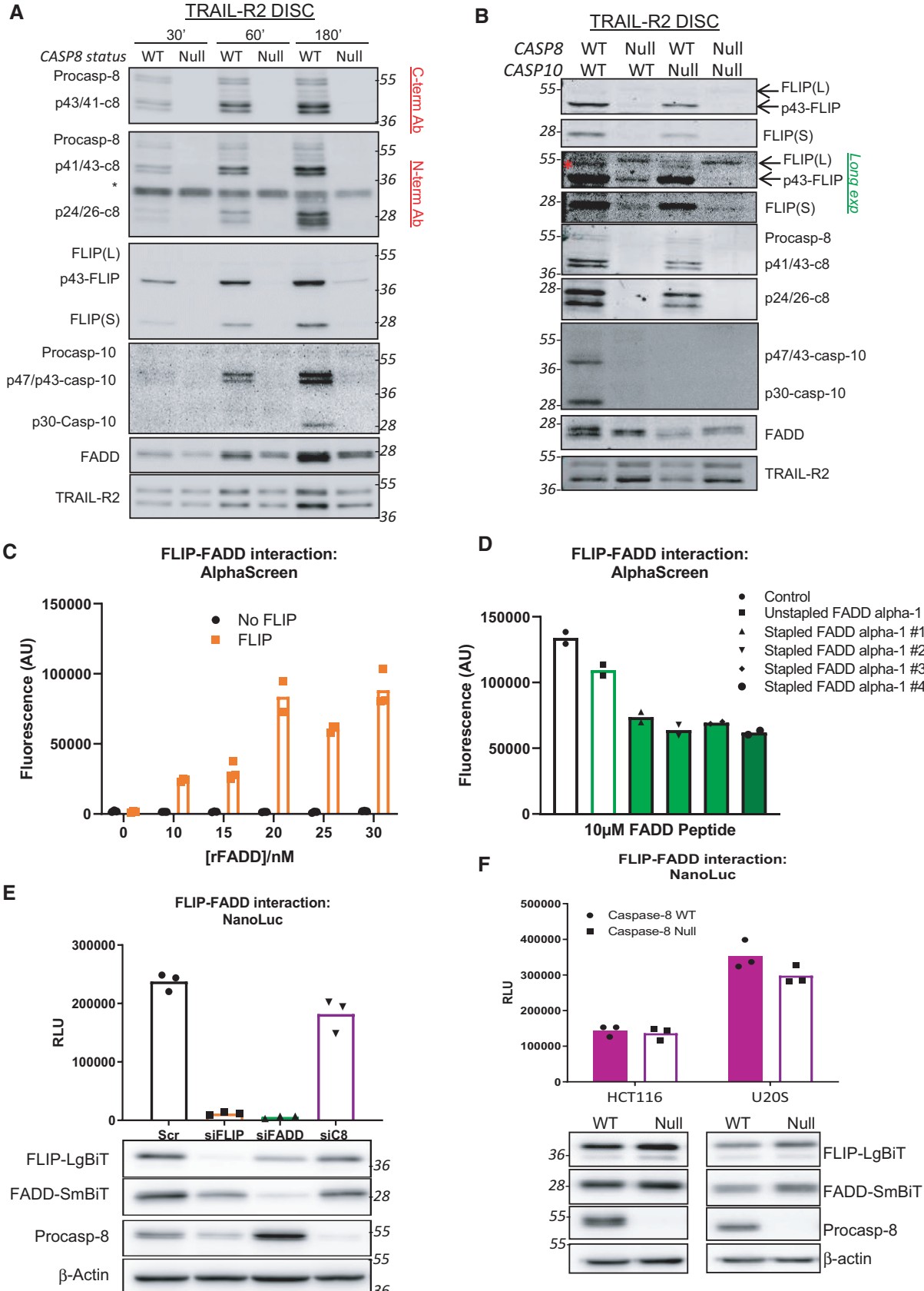

**Figure 3.**

**Figure 3. Although FLIP can interact directly with FADD in a caspase-8-independent manner, its DISC recruitment is highly caspase-8-dependent.**

A  Western Blot analysis of caspase-8, caspase-10, FLIP and FADD recruitment to the TRAIL-R2 DISC in procaspase-8 (WT) or procaspase-8 deficient (Null) A549 cells incubated with AMG655 beads (4×) for either 30, 60 or 180 min. Unbound fractions are shown in Fig EV5A.

B  Western blot analysis of the recruitment of FLIP, caspase-8, caspase-10 and FADD to the TRAIL-R2 DISC in HCT116 parental, caspase-8 null (CASP8), caspase-10 null (CASP10) or CASP 8/10 cells treated with 4× AMG655-conjugated beads for 90 min. Unbound fractions are shown in Fig EV6B. * modified form of FLIP, potentially mono-ubiquitinated p43-FLIP(L).

C  AlphaScreen® assessment of the interaction of recombinant FLIP DED1/2 and recombinant FADD DED.

D  AlphaScreen® assessment of the impact on the FLIP-FADD interaction of unstapled and stapled peptides corresponding to the FADD α1 helix.

E  NanoBiT® assay to quantify FLIP/FADD interactions in a U2OS cell line stably expressing the NanoBiT® constructs, FLIP (LgBiT) and FADD (smBiT) following 48-h silencing of FLIP, FADD or caspase-8. Western blot analysis was used to confirm successful knockdown of target proteins by siRNA.

F  NanoBiT® assay of HCT116 and U2OS wild-type (WT) and procaspase 8 null (Null) cells transiently co-transfected with FLIP (LgBiT) and FADD (SmBiT) NanoBiT® constructs for 24 h. Construct expression was analysed by Western blotting.

Source data are available online for this figure.

however, low amounts were again detectable at the latest timepoint. In isogenic HCT116 models lacking either procaspase-8, procaspase-10, or both, we further confirmed the importance of procaspase-8 for both FLIP and procaspase-10 recruitment (Figs 3B and EV5B). The absence of procaspase-10 had no significant impact on the levels (relative to TRAIL-R2 and FADD) and processing of FLIP and procaspase-8, indicating that its contribution to overall TRAIL-R2 DISC stoichiometry (at least in the HCT116 model) is minimal. It was however notable that in the absence of both procaspase-8 and procaspase-10, only unprocessed FLIP(L) was detected (Fig 3B; *lane 4*), suggesting that the low amounts of FLIP(L) detected in the absence of procaspase-8 are cleaved by procaspase-10 and that these low levels of FLIP can be recruited by direct binding to FADD and/or procaspase-10 (in FADD null cells, no FLIP or caspase-8 is recruited to the TRAIL-R2 DISC, DB Longley, unpublished observations).

To further investigate the direct interaction between the DEDs of FLIP and FADD, we developed a cell-free, fluorescence-based protein–protein interaction assay (AlphaScreen) using recombinant FLIP DED1/2 and the FADD DED. Using this assay, the ability of these proteins to directly interact via their DEDs was confirmed (Fig 3C). Moreover, using recombinant peptides corresponding to the FADD alpha-1 helix, we were able to disrupt the FLIP-FADD interaction in this assay (Fig 3D). The peptides that disrupted the interaction had "hydrocarbon-stapled" backbones to enhance their alpha helicity [31]; the unstapled control peptide had no effect. This demonstrates the importance of the alpha-1 helix of FADD for its interaction with FLIP and the potential for therapeutically targeting this interaction. These results are in agreement with our previous study, which determined the preferential mode of interaction between these 2 proteins to be between the α1/α4 surface of FADD's DED and the α2/α5 surface of FLIP's DED [29].

To determine whether FLIP can also directly interact with FADD in cells, we next developed a cell-based protein–protein interaction system, in which DED1/2 of FLIP (amino acids 1–185) were fused to an 18 kDa fragment of the nanoluciferase enzyme (termed "LgBiT") and full-length FADD was fused to a smaller 1.3kDa fragment of the nanoluciferase enzyme ("SmBiT"). To inhibit FADD DED filament formation (an established artefact of cellular FADD overexpression [32–34], which could potentially affect FLIP binding), we generated an F25A mutation in the FADD-SmBiT fusion; we previously showed that this mutation does not affect FADD's interaction with FLIP in cell-free assays [20]. Because of the low affinity of the LgBiT and SmBiT, the nanoluciferase enzyme is only reconstituted when

the proteins are fused to interact [35]. In cell lines stably expressing the FLIP-LgBiT and FADD(*F25A*)-SmBiT fusions, nanoluciferase activity was detected (Fig 3E). Using siRNAs to deplete each fusion protein, we were able to demonstrate that this interaction was specific and required both binding partners. Furthermore, this interaction was maintained when procaspase-8 was silenced with siRNA. As even small amounts of procaspase-8 can be sufficient to allow DISC formation (Fig 2A), we confirmed that the FLIP and FADD interaction was procaspase-8-independent by transiently expressing the FLIP-LgBiT and FADD-SmBiT fusions in *CASP8* null cells (Fig 3F). Taken together, these results demonstrate that procaspase-8 is required for efficient recruitment of FLIP (and procaspase-10) to the DISC; nonetheless, FLIP can directly interact with FADD via its DEDs in a procaspase-8 (and -10)-independent manner. We also show clearly for the first time that caspase-10 can cleave FLIP(L) at the TRAIL-R2 DISC.

## FADD recruitment to the TRAIL-R2 DISC is impaired in the absence of procaspase-8

In the *CASP8* null A549 model, although FADD was clearly recruited to the DISC, the relative quantities when normalised to TRAIL-R2 were consistently lower compared to the control *CASP8* model (Fig 3A), although not to the same extent as reported for the Fas/CD95 DISC [30]. This effect was observed in multiple *CASP8* null cell lines (Fig 4A–C). Since procaspase-8 and FADD interact via their DEDs, we next used FADD constructs with H9G (on its α1/α4 surface) or F25A (on its α2/α5 surface) substitutions (Fig 4D) to determine the importance of FADD's DED-mediated interactions for its DISC recruitment. As expected, wild-type FADD was efficiently recruited to the DISC; however, despite high levels of expression, neither F25A nor H9G mutant FADD proteins (which contain the death domains (DDs) that mediate its interactions with the receptor's DD) were detectable at the TRAIL-R2 DISC (Fig 4E). As mutation of these surface residues should not affect FADD protein folding, this suggests that the FADD DED is important for its interaction with the DISC, in agreement with an earlier study [36]. Our data suggest that FADD requires DED-mediated interactions on both its α1/α4 and α2/α5 interfaces for recruitment and/or stabilisation at the DISC. To further investigate this, we developed a FADD DED:-caspase-8 DED1/2 NanoLuc system (Fig 4F), which demonstrated significantly reduced affinity of FADD for caspase-8's DEDs when either F25 or H9 are mutated. Collectively, these results are consistent with FADD interacting with procaspase-8 on both its α1/α4 and

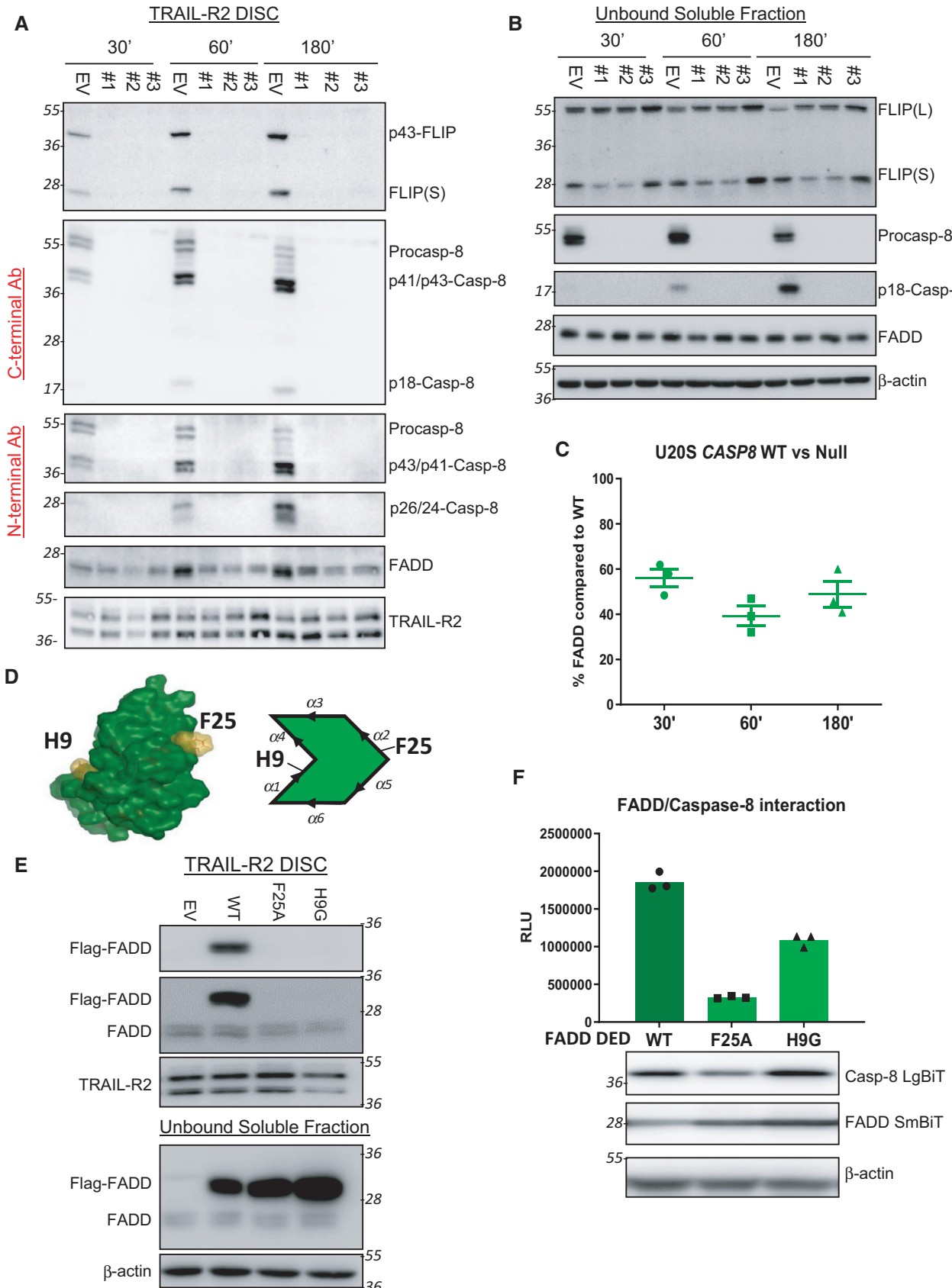

**Figure 4.**

**Figure 4. FADD recruitment to the TRAIL-R2 DISC is impaired in the absence of procaspase-8.**

A Western blot analysis of FLIP, caspase-8 and FADD recruitment to the TRAIL-R2 DISC in U2OS parental cells (EV) and 3 independent caspase-8 null clones (#1, #2, #3) after incubation with AMG655-conjugated beads for 30, 60 or 180 min.
B Western blot analysis of FLIP, caspase-8 and FADD in the soluble unbound fraction from panel (A).
C Quantification of FADD recruitment to the TRAIL-R2 DISCs from (A); FADD levels were normalised to TRAIL-R2 levels in the pull-downs. Densitometry was performed using ImageJ®.
D The structure of FADD was previously published [20]. Here, we present the Connolly (solvent-excluded) surface, with the positions of the H9 and F25 residues highlighted. The accompanying chevron is a short-hand representation of the 6 α-helices of FADD's DED.
E Western blot analysis of FADD recruitment to the TRAIL-R2 DISC in HCT116 cells transiently transfected with an empty vector (EV), Flag-tagged FADD (WT) or Flag-tagged FADD with point mutations at either Phe25 (F25A) or His9 (H9G). Western blotting analysis of the soluble unbound fraction was used to monitor transfection efficiency.
F NanoBiT® assay of U2OS cells transiently co-transfected with caspase 8 (LgBiT) and either WT FADD smBiT or FADD (smBiT) constructs with a point mutation at either phenylalanine 25 (F25A) or histidine 9 (H9G) for 48 h. Western blot analysis was used to assess the expression level of each construct.

Source data are available online for this figure.

α2/α5 surfaces and these interactions being important for FADD binding at the DISC.

### Impact of FLIP deletion on DISC assembly and caspase activation

The above results clearly demonstrate the requirement of procaspase-8 for efficient DISC assembly. We next assessed the importance of FLIP for DISC assembly and processing and activation of procaspase-8. Although siRNA-mediated FLIP(L) down-regulation had little effect on total caspase-8 recruitment in the HCT116 model (in which, as noted above, caspase-8:FLIP(L) DISC ratios are already high), FLIP(L) down-regulation actually inhibited procaspase-8 processing, with accumulation of unprocessed procaspase-8 detected at the TRAIL-R2 DISC (Fig EV5C), which importantly correlated with reduced caspase-8 activity in the unbound fraction at these timepoints (Fig EV5D).

We were unable to generate HCT116, A549 or DU145 FLIP KO models using CRISPR-Cas9, consistent with the FLIP-dependence of these models [37–39]. We therefore obtained a FLIP KO cell line model derived from a HAP1 chronic myeloid leukaemia (CML) model. At early timepoints in the FLIP KO model, although procaspase-8 was recruited, its processing was significantly impaired (Fig 5A), and this correlated with reduced caspase-8 (and executioner caspase-3/7) activity in the unbound fraction (Fig 5B). It was apparent at later timepoints post-TRAIL-R2 DISC activation that it is the kinetics of procaspase-8 processing that is affected by the absence of FLIP(L), as the extent of processing and activation of procaspase-8 at later timepoints was similar to that observed in the parental FLIP-proficient model (Fig 5A and B). This was supported by caspase-8 and caspase-3/7 activity assays on extracts from cells treated for 6 h or 24 h with IZ-TRAIL [40] (Fig 5Ci–ii and Di–ii). Notably, early apoptotic cell death induced by IZ-TRAIL was also significantly attenuated in the *CFLAR* knockout model at 6 h post-treatment (Fig 5Ciii), but was higher at 24 h (Fig 5Diii). These results, which can be attributed to FLIP(L) (this is by far the predominant splice form expressed in this cell line), were further supported by Western blot analysis of PARP cleavage and caspase-8 processing (Fig 5E). Moreover, in 72 h cell viability assays, the *CFLAR* knockout was more sensitive (Fig 5F), despite the initial slower kinetics of procaspase-8 processing. These results indicate that the complete absence of FLIP(L) slows the kinetics of procaspase-8 processing and activation at the DISC, but that ultimately, the amplitude of the apoptotic signal generated is enhanced.

## Discussion

In order to reconcile and integrate the findings described above with our previous findings and the findings of others, we propose a revised model of DISC assembly (Figs 6A and EV5E).

- *DISC Initiation.* Once bound to its trimeric ligand, preformed death receptor trimers are activated [41]. Previous publications have demonstrated a 1:1 ratio of FADD to TRAIL-R2 [42]; therefore, 3 FADDs can potentially be recruited to each activated receptor trimer.
- *FADD's DISC recruitment involves its DED.* Although recruited to the receptor via DD interactions, FADD's DISC recruitment also involves interactions mediated by its DED via both α1/α4 (mediated by H9) and α2/α5 (mediated by F25) surfaces.
- *Inefficient FLIP recruitment in the absence of caspase-8.* Although FLIP can interact directly with FADD in a caspase-8-independent manner, loss of procaspase-8 inhibits FLIP recruitment to the DISC, indicating a critical scaffolding role for procaspase-8 in DISC assembly.
- *Tandem DED chain length is stable.* In contrast to the caspase-8 chain model, our results suggest that the tandem DED chain lengths are remarkably stable across cell lines, extent of TRAIL-R2 activation and levels of FLIP splice form expression; we have depicted a chain length that is consistent with the ratios observed in our experiments.
- *When equi-stoichiometric with caspase-8, FLIP(L) acts as an apoptosis inhibitor.* When procaspase-8 and FLIP(L) are at a 1:1 ratio (Fig 6A *scenario i* and EV5E(iii)), although initial inter-dimer cleavage is rapid, full procaspase-8 processing is significantly inhibited as the number of procaspase-8 homodimers will be low; this will be the scenario when there are low levels of receptor ligation and/or when FLIP expression is elevated (as in many cancers). Unlike the procaspase-8/FLIP(S) heterodimer, the procaspase-8/FLIP(L) heterodimer has catalytic activity and can not only cleave adjacent homo- and heterodimers, but also necroptosis-inducing RIPK1 [18,43,44]. Under these equi-stoichiometric apoptosis-inhibitory conditions, FLIP(L) can therefore also inhibit necroptosis [45].
- *Sub-stoichiometric FLIP(L) accelerates caspase-8 activation.* When the ratio of procaspase-8 to FLIP(L) is greater than 1:1 (Figs 6A *scenario ii* and EV5E(ii)), full activation of procaspase-8 is possible, although not for all recruited procaspase-8 molecules as a

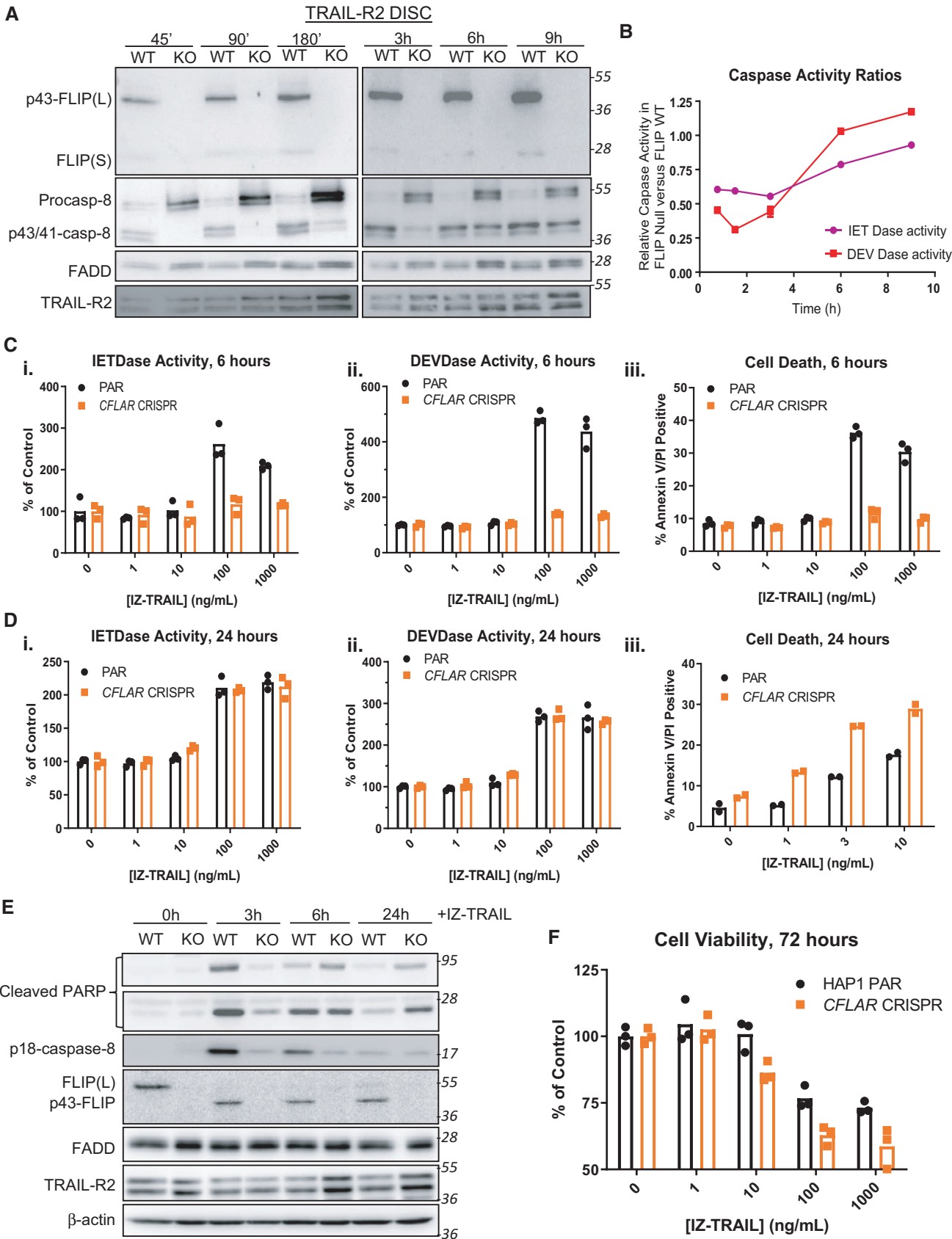

**Figure 5.**

**Figure 5. Impact of FLIP deletion on DISC assembly and caspase activation.**

A   Western blot analysis of FLIP, caspase-8 and FADD recruitment to the TRAIL-R2 DISC in parental (WT) and CFLAR deficient (KO) HAP-1 cells treated with AMG655-conjugated beads for either 45 min (45′), 90 min (90′), 180 min (180′), 3 hours (3 h), 6 hours (6 h) or 9 hours (9 h).
B   Caspase activity assays of the unbound soluble fraction from panel (A).
C   Caspase-8 (i) and caspase-3/7 (ii) activity assays and high content microscopy assessment of cell death (iii) in parental (PAR) and FLIP knockout (CFLAR CRISPR) cells treated with escalating doses of isoleucine zipper TRAIL (IZ-TRAIL) for 6 h.
D   Caspase-8 (i) and caspase-3/7 (ii) activity assays in parental (PAR) and FLIP knockout (CFLAR CRISPR) cells treated with escalating doses of isoleucine zipper TRAIL (IZ-TRAIL) for 24 h. (iii) Flow cytometry assessment of cell death induction in parental (PAR) and FLIP knockout (CFLAR CRISPR) cells treated with IZ-TRAIL for 24 h.
E   Western Blot analysis of Parental (WT) and FLIP knockout (KO) HAP-1 cells treated with 10 ng/ml IZ-TRAIL for 0, 3, 6 or 24 h.
F   Cell viability assay of Parental (PAR) and FLIP knockout (CFLAR CRISPR cells) treated with an escalating dose (0–1,000 ng/ml) of IZ-TRAIL for 72 h.

Source data are available online for this figure.

proportion will be in a heterodimer with FLIP(L). In this scenario, the caspase-8/FLIP(L) heterodimer (but not the caspase-8/FLIP(S) heterodimer; Fig EV5E(iV)) can promote inter-dimer cleavage of adjacent procaspase-8 homodimers (and other heterodimers). As the heterodimer is more efficient at inter-dimer cleavage, which is rate-limiting for caspase-8 activation [11], the presence of low levels of the caspase-8/FLIP(L) heterodimer would be activatory in this scenario, which is why we observed a significant decrease in the kinetics of caspase-8 activation in FLIP(L)-deficient cells (Figs 6A *scenario iii* and EV5E(i)), *i.e.* slower inter-dimer cleavage.

- *Although slower, the extent of cell death is greatest in the absence of FLIP.* In the absence of FLIP(L), the number of procaspase-8 homodimers formed at the DISC will be higher, so although initially slower, the extent of the apoptotic signal can ultimately be greater (Figs 6A *scenario iii* and EV5E(i)); this is why we observed greater cell death and loss of viability at later timepoints in the FLIP KO model than in its FLIP-expressing control.

This model requires interactions between FLIP and procaspase-8's DEDs on both the α1/α4 surfaces of each protein's DED1 and α2/α5 surfaces of each protein's DED2. In a previous study [20], we showed that procaspase-8 preferentially uses F122 on its α2/α5 surface to form homodimers and heterodimers with FLIP in cell-free pull-down studies. Here, we confirm and extend these observations using the NanoLuc system to show that FLIP's interaction with the DEDs of procaspase-8 at the TRAIL-R2 DISC involves both procaspase-8's F122 on its α2/α5 surface and FLIP's F114 on its α2/α5 surface (Fig 6B). Collectively, these results are consistent with caspase-8's DEDs forming bridging interactions as part of a DED chain involving FLIP's DEDs, in which FLIP needs F114 to propagate the chain. Compared to FADD and caspase-8, FLIP appears to be less efficient at forming DED filaments when overexpressed [28]; this may explain why its DISC recruitment is reduced in the absence

of procaspase-8, *i.e.* inefficient formation of homo-oligomeric DED bridging interactions by FLIP. This model also allows for interactions between adjacent DED chains as proposed by a recent study using Cryo-EM in which it was proposed that DED chains can interact at a higher order level and generate intertwining filaments [46]. Interestingly, curvature is predicted in both intra- and inter-molecular DED interactions (Fig 6C). We speculate that there is therefore potential for the DED-containing proteins recruited to the DISC to form structures such as those depicted in Fig 6D, with the basic functional unit consisting of DED-mediated bridging interactions between adjacent TRAIL-R2 trimers. This would be consistent with the hexameric nature of the highly potent 2nd generation TRAIL-R2 agonists such as MEDI3039 (Fig EV2D) that are in early clinical trials [27] and late pre-clinical development [25,47].

In agreement with the literature, our model suggests that FLIP(S) is the more potent inhibitor of procaspase-8 activation at the DISC, as a 1:3 ratio of FLIP(S):procaspase-8 would be theoretically sufficient to block caspase-8 processing (Fig EV5E(iv)), whereas FLIP(L) needs to be present at equi-stoichiometric levels to inhibit caspase-8 processing (Fig EV5E(iii)). Our data and model also argue against FLIP(S) acting as a caspase-8 DED chain terminator as has been proposed [28]. To confirm this, we performed size-exclusion chromatography of lysates from IZ-TRAIL-treated cells overexpressing FLIP(S); notably in agreement with our proposed model, we isolated FLIP(S)-containing complexes in fractions of 2 MDa and above (Fig 6E). Endogenous FLIP(S) and p43-FLIP(L) were also detectable in the 2 MDa fraction. If FLIP(S) acts as a chain terminator at the DISC, it would be expected to be confined to significantly lower MW fractions (< 1 MDa).

In concluding, our data suggest that at low levels of TRAIL-R2 activation, when there are similar levels of procaspase-8 and FLIP(L) recruited to the DISC, there will be a predominance of

**Figure 6. Proposal of a novel DISC assembly model.**

A   Schematic representation of a model for DISC structure consistent with the experimental observations in this paper. Key: green chevrons: FADD DED; orange chevrons: FLIP DED1/2; purple chevrons: caspase-8 DED1/2; black triangle: trimeric TRAIL-R2.
B   NanoBiT® assay performed on TRAIL-R2 DISC IPs (90′) carried out in U2OS caspase-8 null cells co-transfected with wild-type or F114A FLIP-LgBiT fusion proteins and wild-type or F122A caspase-8-SmBiT fusion proteins.
C   Docking Connolly surface models of interactions between the DEDs of FADD (green), FLIP (orange) and caspase-8 (purple). The structures of the FADD, FLIP and caspase-8 DEDs were previously published [20].
D   Potential DISC structure linking two receptor trimers via DED chains. Putative caspase domain interactions are depicted.
E   Western blot analysis of size-exclusion chromatography experiments in which HCT116 cells transfected with Flag-tagged wild-type (WT) FLIP(S) were treated with 100 ng/ml IZ-TRAIL for 3 h prior to cell lysis and separation of low and high MW complexes using a size-exclusion column. The fractions eluted at 2 MDa and > 2 MDa are presented.

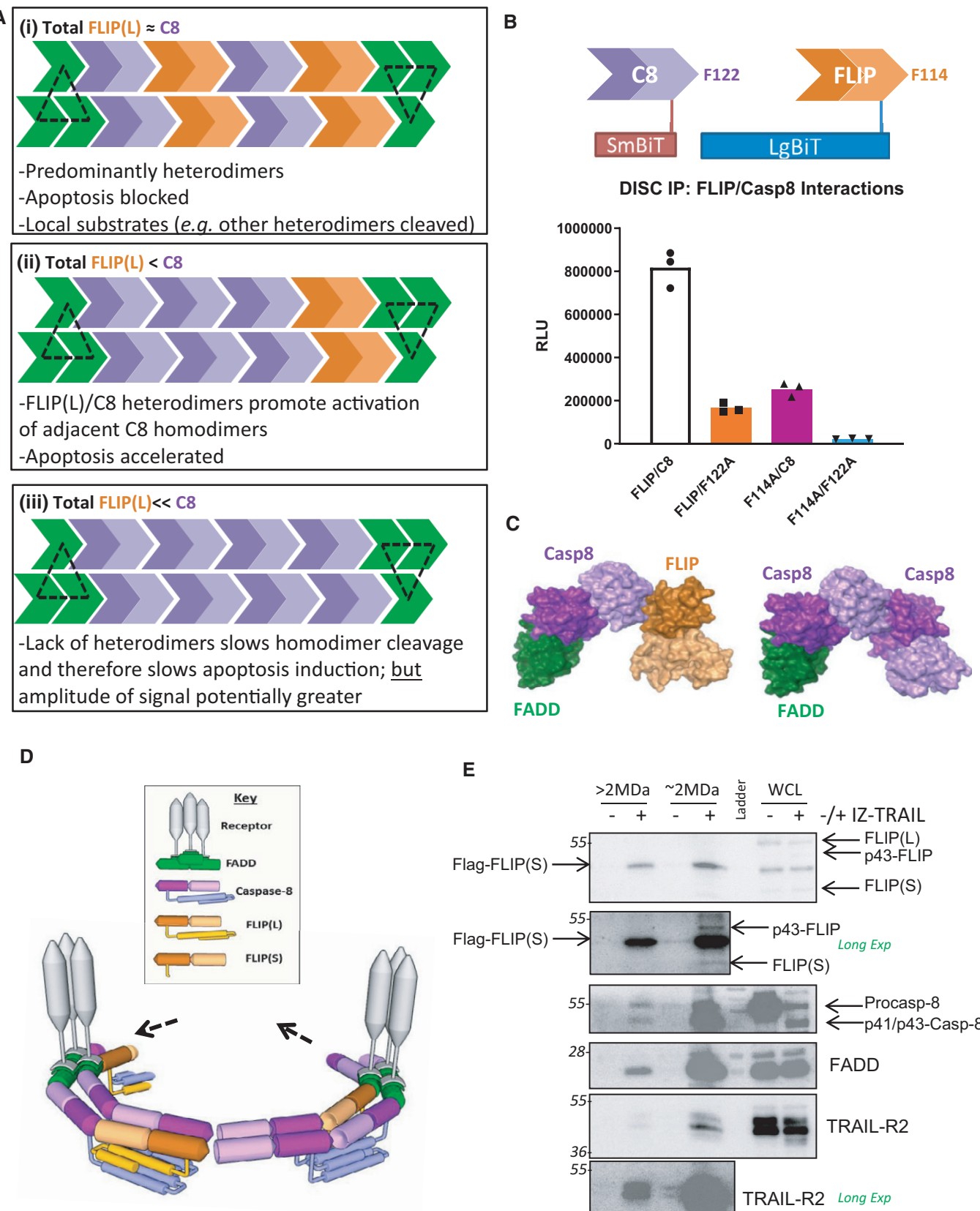

Figure 6.

heterodimers, and apoptosis will be blocked due to formation of insufficient numbers of procaspase-8 homodimers. In addition, due to the enzymatic activity of the heterodimer, co-recruited RIPK1 would be cleaved, thereby preventing necroptosis. The prevention of necroptosis may explain the high FLIP(L) expression and retention of caspase-8 expression observed in many cancers. At higher levels of receptor engagement, the caspase-8:FLIP(L) ratio will increase as total cellular levels of FLIP(L) are typically lower level than procaspase-8 (even when its expression is elevated in tumour cells) [22] and the heterodimer will then catalyse the processing of adjacent homodimers, thereby accelerating procaspase-8 activation and apoptosis induction. Therefore, therapeutic agents that cause relatively small changes in FLIP(L)'s DISC recruitment may have profound effects on cell death signalling by tipping DISC stoichiometry in favour of rapid caspase-8 activation and apoptosis. In this regard, the stapled peptides based on the FADD α1 helix (Fig 3D) show that disrupting the FLIP-FADD protein–protein interaction is possible. However, any novel therapeutic targeting the FLIP-FADD interaction would have to exhibit selectivity for FLIP-FADD over procaspase-8:FADD; the first small molecules that do this and thereby enhance TRAIL-induced caspase-8 activation and apoptosis induction have been identified [48].

# Materials and Methods

### Materials

Antibodies were sourced as follows: AMG655 (conatumumab) was sourced from Amgen Inc (Thousand Oaks, CA, USA); FADD mouse monoclonal antibody was from BD transduction laboratories (catalog #556402); N-terminal specific procaspase 8 antibody was purchased from Abcam (catalog #ab32125); C-terminal specific procaspase-8 antibody was purchased from Enzo® life sciences (catalog # ALX-804-242-C100); FLIP antibody (NF6; catalog # AG-20B-0056-C100) was purchased from Adipogen; TRAIL-R2 was purchased from Cell signalling technology (catalog #3696); and procaspase-10 was purchased from MBL international (catalog #M059-3). Horseradish peroxidase-conjugated secondary antibodies were purchased from cell signalling technology; LI-COR mouse (IRDye® 800CW) and rabbit (IRDye® 680CW) secondary antibodies were purchased from LI-COR. MS-275 (entinostat) was obtained from Selleck Chemicals (Newmarket, UK); annexin V-FITC was obtained from BD biosciences; and z-Val-Ala-Asp(OME)-FMK (zVAD-FMK) was purchased from Sigma-Aldrich (Gillingham, UK). Transfections were completed using FuGENE® HD transfection reagent (Promega, UK). The Dynabead® antibody coupling kit and a Dynamag™-2 magnetic rack were obtained from Life Technologies (Paisley, UK). FADD peptides were generated by Almac Sciences (Edinburgh, UK).

### Cell culture

U20S, A549, HCT116, DU145 and PC3 cells were obtained from American Type Culture Collection (ATCC, Manassas, VA, USA). U20S, DU145 and PC3 cells were cultured in RPMI 1640 (Sigma, UK) supplemented with 10% FCS (Thermo Fisher, UK). All HCT116 cells were cultured in McCoys 5A (modified) medium (Thermo Fisher, Paisley, UK) supplemented with 10% FCS and 1% sodium

pyruvate. HAP1 cells were obtained from Horizon Discovery (Cambridge, UK). Immediately after purchase, early passage stocks of each cell line were frozen down. After thawing, cells were kept for a limited number of passages and were regularly screened for the presence of mycoplasma using the MycoAlert Mycoplasma Detection Kit (Lonza, Basel, Switzerland).

### Expression constructs

pNBE1, pNBE2 and pNBE4 constructs were obtained from Promega; FLIP, FADD and caspase-8 sequences were cloned in using relevant primers (Eurofins Genomics, Germany) and the KOD Hot start DNA polymerase kit (Merck, Darmstadt, Germany). Site-directed mutagenesis was completed with the same kits. DNA sequencing was carried out by GATC (Eurofins Genomics, Germany).

### TRAIL-R2 DISC immunoprecipitation (IP) assay

Performed as previously described [20]. Briefly, 75 μl of Dynabeads (Thermo Fisher, UK) coated with 5 μg (4×), 2.5 μg (2×) or 1.25 μg (1×) of AMG655 was added to $5 \times 10^6$ cells for the indicated time. Cells were then lysed with DISC IP buffer (0.2% NP-40, 20 mM Tris–HCL (pH 7.4), 150 mM NaCl and 10% glycerol) for 30 min and subsequently washed five times in DISC IP buffer. Recruitment of DISC proteins was assessed by Western blotting.

### NanoBiT® assay

U20S cells stably or transiently co-expressing LgBiT and SmBiT fusion proteins were seeded at $5 \times 10^3$ in a white-walled 96-well plate. The following day, media were aspirated from the cells and replaced with 40 μl of Opti-MEM® mixed with NanoGlo® Luciferase Assay Reagent (160× dilution). This was incubated at room temperature for 5 min, protected from ambient light, after which luminescence was read on a Synergy 4 plate reader (Biotek, USA) at a sensitivity of 200. Blank readings were taken in all cases to account for background luminescence.

### Caspase activity assays

*From cell lysates*: 25 μl of Caspase-Glo®-3/7 or -8 reagent (Promega, Southampton, UK) was added to 5μg (caspase-3/7 activity) or 10μg (caspase-8 activity) of cell lysate, made up to 25 μl with PBS in a white-walled 96-well plate. Each sample was plated in triplicate. The plate was incubated in the dark at room temperature for 45 min with gentle agitation. After incubation, the plate was read on a Synergy 4 plate reader (Biotek, USA) at a sensitivity of 200. Blank readings were taken in all cases to account for background luminescence. *In live cells*: Cells were seeded in triplicate at $5 \times 10^3$ per well in a white-walled 96-well plate. All drugging was performed in a total volume of 50 μl. After experimental procedures were completed, 50 μl Caspase-Glo® reagent (Promega, Southampton, UK) was added to each well and the plate covered in foil to protect the reagent from ambient light. The plate was then incubated at room temperature for 45 min with gentle agitation. After incubation, the plate was read on a Synergy 4 plate reader (Biotek, USA) at a sensitivity of 200. Blank readings were taken in all cases to account for background luminescence.

## Cell viability assay

Cell viability was assessed by CellTiter-Glo® assay. Cells were seeded at $1.25 \times 10^3$ in white-walled 96-well plates. The following day, media were aspirated and media replaced with 200 µl of growth medium containing the relevant concentration of drug. Once experimental parameters were complete, 150 µl of media was removed from the well and 50 µl of CellTiter-Glo® reagent added to the samples. The plates were incubated at room temperature for 10 min with gentle agitation and protected from ambient light. Luminescence was then read on a Synergy 4 plate reader (Biotek, USA) at a sensitivity of 200.

## Generation of CRISPR-Cas9 KO models

Caspase-8 null cells were generated by lentiviral infection with pLentiCRISPRV2 with the gRNA: AAGTGAGCAGATCAGAATTG, which was provided as a kind gift from Prof. Galit Lahav [49]. Stable lines were selected in puromycin.

## Western blotting and densitometry

Western blotting was carried out using a G-Box digital developer (Syngene, Cambridge, UK) to develop chemiluminescent blots and a LI-COR® Odyssey for infrared imaging. Densitometry for chemiluminescent blots was carried out on ImageJ®. The LI-COR® Odyssey Infrared Imaging System (version 3.0.30) was used to quantify bands for all quantitative DISC IPs.

## Flow cytometry

Annexin V/Propidium iodide flow cytometry was carried out on a BD LSR-II flow cytometer (BD Biosciences, San Diego, CA, USA) using FITC-Tagged Annexin V antibody (BD Biosciences) and propidium iodide (Sigma-Aldrich).

## High content microscopy

3,000 cells per well were seeded onto black 96-well glass-bottom plates (Cellvis, Mountain View, CA) in a volume of 100 µL/well prior to treatment. After treatment, 50µL of cell staining solution (1 µg/ml propidium iodide (PI) plus 30 µg/ml Hoechst 33342 in PBS) was added to each well. Plates were then incubated in the dark for 15 min at room temperature prior to reading on an ArrayScan XTI Live High Content Platform (Thermo Fisher Scientific, Waltham, MA). Hoechst nuclear stain was used to locate single cells prior to measuring individual fluorescence intensities for PI. Control wells containing positive/negative populations for apoptosis were used to set a threshold for scoring individual cells as "live" or "dead" based on average PI fluorescence intensity. This binary score was applied to the entire plate to generate a readout of % PI-positive cells per well from a total of 2,000 assessed cells.

## siRNA transfections

All siRNAs (SCR, FLIP, C8, FADD) were obtained from Dharmacon (Chicago, IL, USA), and transfections were carried out using Lipofectamine RNA iMAX (Life Technologies, Paisley, UK) as previously described [38].

## Sucrose gradients

HCT116 Bax null cells were seeded at $10 \times 10^6$ in a P140 and left overnight. The following day, cells were transfected with 3µg of Flag-tagged FLIP(S) and incubated at 37°C for 24 h. Following this, the cells were treated with 100 ng/ml of IZ-TRAIL for 3 h and lysed in 1.5 ml ice cold sucrose buffer (5% (w/v) sucrose, 20 mM HEPES, 120 mM NaCl, 2 mM EDTA, 2 mM KCl and 1% CHAPS) supplemented with a protease inhibitor cocktail (EMD Millipore, Germany) for 2 h on ice. Lysates were the filtered using a Whatman™ 13 mm Sterile Polyethersulfone Syringe Filter (0.2 µM, Whatman/GE Healthcare, UK). Lysates were then separated by size-exclusion chromatography with a Superose™ 6 10/30O GL column connected to an AKTA Purifier protein purification system (GE Healthcare) and separated at a flow rate of 0.3 ml/min into 0.5 ml fractions. For SDS–PAGE analysis, aliquots of 100 µl of each fraction were taken, and the presence of proteins was assessed by Western blotting.

## AlphaScreen® assay

The Amplified Luminescent Proximity Homogenous Assay (Perkin Elmer) for studying the biomolecular interactions of protein partners in a microplate format. The AlphaScreen assay was carried out using recombinant FLIP DED1/2 (amino acids 1–184) with a C-terminal His-tag and FADD DED (amino acids 1–90) with an N-terminal GST-tag in a total volume of 25 µl in a 384-well Optiplate plate (Perkin Elmer). All dilutions and incubations were carried out in 25 mM Tris, pH 7.5, 300 mM NaCl, 0.0007% Tween-20 and 0.05% BSA buffer. FLIP and FADD were co-incubated for 30 min at room temperature prior to addition of 5 µl of nickel chelate acceptor beads (diluted 1/50 in assay buffer) and 5 µl of assay buffer for 45 min at room temperature. After this, 5 µl of glutathione donor beads (diluted 1/50 in assay buffer) was then added and incubated for a further 45 min at room temperature with gentle agitation. Fluorescence was then assessed using a Synergy 4 microplate reader (Biotek) according to the manufacturer's instructions (excitation 680/30 nm, emission 570/100 nm using a top dichroic mirror 635 nm).

## Molecular modelling

Molecular modelling was performed using published structures as previously described [20].

Expanded View for this article is available online.

## Acknowledgements

This work was funded by grants from The Wellcome Trust (110371/Z/15/Z), Cancer Research UK (C11884/A24387), Northern Ireland Department for the Economy (NI DfE) (SFI-DEL 14/1A/2582) and a NI DfE studentship (LMH). We thank Prof Henning Walczak (UCL Cancer Inst) for supplying the IZ-TRAIL expression construct.

## Author contributions

LMH: conceptualisation, methodology, validation, formal analysis, investigation, writing (original draft) and visualisation; JPF: investigation; CAH:

methodology, investigation and supervision; JM: methodology and investigation; KM: investigation; CM: investigation; JZR: investigation; TS: investigation and visualisation; NC: resources and supervision; SSM: funding acquisition, writing (review & editing); CJS: funding acquisition, writing (review & editing); TH: funding acquisition, writing (review & editing): DBL: conceptualisation, methodology, validation, formal analysis, writing (original draft), visualisation, supervision, project administration and funding acquisition.

## Conflict of interest

The authors declare that they have no conflict of interest.

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
