## [Review Process File · EMBO Reports]

A revised model of TRAIL-R2 DISC assembly explains how FLIP(L) can inhibit or promote apoptosis

L Humphreys, J Fox, C Higgins, J Majkut, T Sessler, K McLaughlin, C McCann, JZ Roberts, N Crawford, SS McDade, CJ Scott, T Harrison, DB Longley

Review timeline:

Submission date:	10 September 2019
Editorial Decision:	23 October 2019
Revision received:	16 November 2019
Editorial Decision:	17 December 2019
Revision received:	20 December 2019
Accepted:	13 January 2020

Transaction Report:

1st Editorial Decision

23 October 2019

Thank you for the submission of your research manuscript to our journal. I apologize for the delay in handling your manuscript but we have only now received the full set of referee reports that is copied below.

As you will see, the referees acknowledge that the findings are potentially interesting. However, referees 1 and 2 also point out several concerns and have a number of suggestions for how the study and its conclusions should be strengthened, and I think that all of them should be addressed.

Given these constructive comments, we would like to invite you to revise your manuscript with the understanding that the referee concerns (as detailed above and in their reports) must be fully addressed and their suggestions taken on board. Please address all referee concerns in a complete point-by-point response. Acceptance of the manuscript will depend on a positive outcome of a second round of review. It is EMBO reports policy to allow a single round of revision only and acceptance or rejection of the manuscript will therefore depend on the completeness of your responses included in the next, final version of the manuscript.

Revised manuscripts should be submitted within three months of a request for revision; they will otherwise be treated as new submissions. Please contact us if a 3-months time frame is not sufficient for the revisions so that we can discuss the revisions further.

2) individual production quality figure files as .eps, .tif, .jpg (one file per figure).

Please download our Figure Preparation Guidelines (figure preparation pdf) from our Author Guidelines pages
<https://www.embopress.org/page/journal/14693178/authorguide> for more info on how to prepare your figures.

4) a complete author checklist, which you can download from our author guidelines (<<https://www.embopress.org/page/journal/14693178/authorguide>>). Please insert information in the checklist that is also reflected in the manuscript. The completed author checklist will also be part of the RPF.

5) Please note that all corresponding authors are required to supply an ORCID ID for their name upon submission of a revised manuscript (<<https://orcid.org/>>). Please find instructions on how to link your ORCID ID to your account in our manuscript tracking system in our Author guidelines (<<https://www.embopress.org/page/journal/14693178/authorguide#authorshipguidelines>>)

6) We replaced Supplementary Information with Expanded View (EV) Figures and Tables that are collapsible/expandable online. A maximum of 5 EV Figures can be typeset. EV Figures should be cited as 'Figure EV1, Figure EV2" etc... in the text and their respective legends should be included in the main text after the legends of regular figures.

7) We would also encourage you to include the source data for figure panels that show essential data, such as uncropped Western blot data.

Numerical data should be provided as individual .xls or .csv files (including a tab describing the data). For blots or microscopy, uncropped images should be submitted (using a zip archive if multiple images need to be supplied for one panel). Additional information on source data and instruction on how to label the files are available <<https://www.embopress.org/page/journal/14693178/authorguide#sourcedata>>.

8) IMPORTANT information regarding figure legends and statistics:

- Please specify the name of the statistical test used to generate error bars and P values, the number (n) of independent experiments underlying each data point (not replicate measures of one sample), and the test used to calculate p-values in each figure legend. Discussion of statistical methodology can be reported in the materials and methods section, but figure legends should contain a basic description of n, P and the test applied.

IMPORTANT: Please note that error bars and statistical comparisons may only be applied to data obtained from at least three independent biological replicates. If the data rely on a smaller number of biological replicates or on technical replicates, scatter blots showing individual data points must be shown.

- Graphs must include a description of the bars and the error bars (s.d., s.e.m.).

9) METHODS: please note that we have no Word limit for the methods part of a manuscript. I would therefore encourage you to replace the reference to "performed as previously described", page 18, line 419 with at least some minimal information about the conditions used for the TRAIL-R2 DISC IP assay.

10) Our journal encourages inclusion of *data citations in the reference list* to directly cite datasets that were re-used and obtained from public databases. Data citations in the article text are distinct from normal bibliographical citations and should directly link to the database records from which the data can be accessed. In the main text, data citations are formatted as follows: "Data ref: Smith et al, 2001" or "Data ref: NCBI Sequence Read Archive PRJNA342805, 2017". In the Reference list, data citations must be labeled with "[DATASET]". A data reference must provide the database name, accession number/identifiers and a resolvable link to the landing page from which the data can be accessed at the end of the reference. Further instructions are available at <<https://www.embopress.org/page/journal/14693178/authorguide#referencesformat>>.

11) As part of the EMBO publication's Transparent Editorial Process, EMBO reports publishes online a Review Process File to accompany accepted manuscripts. This File will be published in conjunction with your paper and will include the referee reports, your point-by-point response and all pertinent correspondence relating to the manuscript.

I look forward to seeing a revised version of your manuscript when it is ready. Please let me know if you have questions or comments regarding the revision.

REFeree REPORTS

Referee #1:

Humphreys et al. investigate the initial steps of death receptor signaling, a subgroup of TNF receptors that is able to induce apoptosis. They concentrate on the death receptor TRAIL-R2. The TRAIL system has been shown to kill preferentially tumor cells. Yet, some cancer cells are resistant and therefore, a deeper understanding of the molecular mechanisms of death receptor signaling is warranted in order to develop novel anti-cancer therapies.

Initially, the authors analyze the stoichiometry of the TRAIL-R2 DISC using quantitative immunoblotting and novel, improved standards. They find a molecular ratio of the adapter protein FADD to the tandem DED-containing proteins (caspase-8, c-FLIP) of about 1:3, which is in between the number they published previously and the numbers published by others. The model derived from this data suggests a DISC that is composed of a preformed receptor trimer that recruits 3 FADD molecules in total. Each FADD molecule recruits (on average) 3 tandem DD proteins into the receptor complex. Of note, this stoichiometry can explain the hexameric nature of efficient death ligands that have been described for the TRAIL (Morgan-Lappe et al., 2017) and CD95 systems (Holler et al., 2003).

Two other important findings are based on CRISPR knockout cell lines. First, they are able to show that c-FLIP can be recruited to FADD directly, i.e. in the absence of caspase-8. This is important because studies using an in vitro reconstituted DISC concluded that caspase-8 is required for DISC recruitment of c-FLIP. However, the data by Humphreys et al. show that this is not the case in living cells. Secondly, the authors show that in the absence of c-FLIP long activation of caspase-8 is delayed. These findings help to explain the dual role (anti- and pro-apoptotic) of c-FLIP in death receptor signaling that is described in the literature.

In summary, Humphreys et al. report very significant findings that improve our understanding of death receptor signaling and have important implications for TRAIL-based anti-cancer therapies. However, in light of the controversial nature of some of their findings a couple of points need to be addressed and clarified.

Page 3: It is stated that the three cell lines used in this study represent cellular models for TRAIL resistance and sensitivity. However, hardly any cell death assays are done in the current study. The authors need to show the cell death rates (Annexin V, PI exclusion or similar) of the cell models employed with the TRAIL reagents used in the study.

Page 4: The ratios of FADD to tandem DED proteins identified by the authors in this study (1:3) and previously (1:2) differs partially (1:3 to 1:5 Schleich et al., 2012) and substantially from the findings by others (1:9 Dickens et al., 2012). How do the authors explain these differences? Is it a matter of sensitivity (immunoblotting vs. mass spectrometry) or a matter of different cell lines used or something else? The authors should provide potential explanations.

Page 6, line 133: The authors state that the ratio for caspase-8:c-FLIP is about 7:1. This is true for the HCT116 cell line. However, in A549 and DU145 cells, the ratio is much smaller. Please clarify.

Page 6, line 135: The authors conclude that a 1:1 ratio between caspase-8 and c-FLIP is required to inhibit death receptor-mediated apoptosis. However, the c-FLIP levels are much lower than caspase-8 in many cell types (e.g. this study; Dickens et al., 2012; Scaffidi et al., 1999; Schleich et al., 2012). How do the authors envisage that c-FLIP can block cell death at all? After all, c-FLIP can inhibit death receptor mediated apoptosis in cancer cells even when it is lower expressed than in normal tissue (Ewald et al., 2011).

Figure 2: The calculation of the caspase-8:c-FLIP ratio is missing in the siRNA experiments. Please provide these calculations. Second, I am not convinced that the band marked with an asterisk on panel E is ubiquitinated p43-c-FLIPlong. The double band could also be endogenous and Flag-tagged c-FLIPlong (full length). Please provide data for ubiquitination or rephrase. Third, the phrase "...followed by a 90 minute DISC IP." (panel E) is misleading. Was the IP performed for 90 minutes or were the cells stimulated for 90 minutes with anti-TRAIL-R2 antibody followed by a DISC IP? Please rephrase accordingly.

Figure 5, panel B: What does "FLIP KO/FLIP WT" mean (labeling y-axis)? If this is the simple ratio, then the activity of caspase-8 in c-FLIP knockout cells does not reach the activity level of wildtype cells; not even after 10 hours of stimulation. If so, how are higher caspase-3/7 activity levels reached at later time points? Please explain!

Page 16, line 341: The authors cite a paper stating that c-FLIP does not form DED filaments. However, looking at the data in the cited paper, it is obvious that c-FLIP overexpression results in aggregates that are visible as GFP speckles (though they do not form as extensive filaments as caspase-8 DEDs). This is consistent with another study showing that isolated DEDs of murine c-FLIP form filaments and that murine c-FLIP enhances FADD filaments in contrast to viral FLIP, which actually inhibits FADD DED filament formation (Ueffing et al., 2008). Hence, a putative inefficient oligomerization of c-FLIP might not be the explanation for reduced DISC recruitment in the absence of caspase-8. The authors might want to revisit the discussion.

Page 16, line 359/360: The authors state that endogenous c-FLIP is detectable in the 2 MDa fraction. However, this is hardly visible. Please improve the data.

Page 16, line 362: "...our findings resolve the enigmatic role of FLIP(L) at the DISC." While the authors provide strong data on the dual role of c-FLIPlong in cells, explanations for pro cell death functions of c-FLIPlong have been published before (e.g. Boatright et al., 2004; Fricker et al., 2010). Therefore, the statement should be toned down.

Minor points

The authors give the concentration of the anti-TRAIL-R2 antibody as 1x, 2x and so on. Real

concentrations should be given a allow for reproducibility by other investigators.

Page 6, line 134: Figure 1 should read Figure 1E.

Figure 4: Please include a description of the DED structure model including the H9 and F25 residues in the figure legend (panel D).

Page 12, line 276: "long-term viability assays" I would reserve this term for clonogenic assays (or similar) that run for 1-3 weeks. Here, cells were analyzed after 72 hours. Please rephrase.

Figure 5, panel Diii: What does "% apoptosis" mean? Which assay has been performed? Is it Annexin V/PI as in Figure 5Ciii? Please clarify.

Page 14, line 291: I suggest to write "... each activated receptor trimer." To make it clearer for non-specialist readers.

Page 14, line 310/311: Please define "these conditions". After all, the manuscript discusses that c-FLIP_{long} can have pro- as well as anti-apoptotic functions. Again, this will help non-specialist readers to follow the story.

Referee #2:

Humphreys et al dissected the relationship between caspase-8 and FLIP in the TRAIL-R2 DISC. They used multiple independent approaches to formulate a detailed model describing the effect of various ratios of FLIP to caspase-8 on the protease's activation status. I feel their data help to clarify some formerly confusing aspects of death receptor signaling. My comments below are intended to further improve a carefully-designed and well-conducted study.

Specific comments

Introduction: Prior data pertaining to the impact of FLIP on caspase-8 activity is well described. I feel it would be useful to also tell/remind readers about FLIP's ability to modulate caspase-8 specificity, as described by Pop et al, 2011 (PMID: 21235526).

Fig 1 A: Were the images showing caspase-8 p24/26 from the same blots as those showing pro-casp-8 and the p41/43 species? If so, why are they shown in separate boxes, cropping out the region between 26 and 41kDa? The positions of molecular weight markers should be shown on all blots. This will help readers consider whether, for example, the apparent lack of a FLIP(L) band in the DISC blot really reflects the absence of this protein in the DISC, or just cropping of the image below the band.

In Figure 2, the concentration of the scrambled siRNA should be stated in the legend or labeled in the figure. Why did this treatment boost caspase activity, alter the TRAIL-R2 band pattern (splicing?), and promote FLIP cleavage?

Fig 2D,G, 5C: How confident are the authors that IETDase activity reflects caspase-8 activity? Caspase-3 cleaves this sequence more efficiently than caspase-8 (PMID: 17975551).

Fig 3A, B/Supp Fig 6, text lines 170-174: Why is less caspase-10 recruited to the DISC when caspase-8 is absent, despite the cells expressing normal levels? The authors note this but do not comment on its significance. Does this imply the existence of caspase-8/10 mixed dimers?

Line 161. I feel the subtitle "FLIP can interact directly with FADD in a caspase-8-independent manner" is a bit misleading. To my eyes, the main finding from Fig 3 is that FLIP recruitment to the DISC is dramatically reduced when caspase-8 (+/- 10) are absent. Yes, a tiny amount is still recruited in cells lacking these caspases, so I guess FLIP CAN be recruited in a caspase-8-independent manner, but that subtitle may give the erroneous impression that recruitment of FLIP is unaffected by caspase-8 removal.

Lines 178-180, re Fig 3B: "low amounts of FLIP(L) detected in the absence of procaspase-8 are

cleaved by procaspase-10 and that these low levels of FLIP can be recruited by direct binding to FADD". Why would this recruitment of FLIP to the DISC in caspase-8-null cells have to be via direct interaction with FADD? Couldn't recruitment of those few FLIP molecules be via binding to the few caspase-10 molecules at the DISC, which are (as the authors note) presumably responsible for their cleavage?

Fig 3A/B versus E/F: Why does the presence of caspase-8 make a massive difference to FLIP recruitment to the DISC, but has only marginal impact on the ability of FLIP and FADD to interact in the nanoluciferase system? Is this because the nanoluciferase assay reports on interactions of soluble cytosolic proteins, but the DISC IPs only analyze binding events at ligated TRAIL-R2? Does this provide some clues about possible conformational differences between DISC-associated FADD/FLIP and the soluble cytosolic versions of those proteins? Alternatively, could the fusion of the luciferase domains to the proteins affect their folding? I think readers would appreciate the authors' explanations for the differential requirement for caspase-8, for FLIP/FADD binding in these different assays.

Line 237: "neither F25A nor H9G the mutant FADD proteins...". Should "the" be removed?

Fig 4D/E, lines 234-244: Have the structures of the F25 and/or H9 FADD mutants been determined? Has it been confirmed that these mutations do not affect the folding of the DD?

Legend to Figure 4D: The structure should be described (and referenced)

Line 310. I didn't notice any data regarding RIPK1 processing, so presumably this statement was based on prior demonstrations of RIPK1 cleavage by caspase-8-FLIP dimers. If so, the relevant papers should be cited.

Lines 309-311, Fig 6A "procaspase-8/FLIP(L) heterodimer has catalytic activity and can not only cleave adjacent homo- and hetero-dimers, but also necroptosis-inducing RIPK1. Under these conditions, FLIP(L) is therefore potentially both anti-apoptotic...". Why is the catalytic activity of the caspase-8/FLIP heterodimer characterized as anti-apoptotic? I understand that heterodimers are less pro-apoptotic than caspase-8 homodimers, but they're still pro-apoptotic, aren't they? According to Pop et al, the heterodimer processes procaspase-3 around half as efficiently as a caspase-8/caspase-8 homodimer. Fig 2 F/G showed that cells whose DISCs contain slightly more FLIP than caspase-8 still contain about a third as much caspase activity as cells with a higher ratio of caspase-8 to FLIP.

1st Revision - authors' response

16 November 2019

Referee #1:

Major Points

Page 3: It is stated that the three cell lines used in this study represent cellular models for TRAIL resistance and sensitivity. However, hardly any cell death assays are done in the current study. The authors need to show the cell death rates (Annexin V, PI exclusion or similar) of the cell models employed with the TRAIL reagents used in the study. Additional data are now presented in Supplementary Figure 2 to address this, with additional data using a multivalent form of recombinant TRAIL (isoleucine zipper (IZ)-TRAIL) and a 2nd generation TRAIL-R2-selective agonist MEDI3039 also presented.

Page 4: The ratios of FADD to tandem DED proteins identified by the authors in this study (1:3) and previously (1:2) differs partially (1:3 to 1:5 Schleich et al., 2012) and substantially from the findings by others (1:9 Dickens et al., 2012). How do the authors explain these differences? Is it a matter of sensitivity (immunoblotting vs. mass spectrometry) or a matter of different cell lines used or something else? The authors should provide potential explanations. As the Reviewer points out, the ~1:3 ratio (ranging from approximately 1:2 to 1:4; see Supplementary Figure 4D) that we obtained in this study is not that different to the ratios obtained by Schleich *et al.*, which were ~1:5 by

quantitative immunoblotting and ~1:2 by mass spec analyses. Moreover, Dickens *et al.* reported ratios of 1:3, 1:4 and 1:4.5 in Z138, BJAB and Jurkat cells respectively by mass spec; again, not that different from the ratios we obtained in this study. Thus, overall, despite using different methods and different cell line models (we used epithelial models whereas the other cited studies used haematological models), the results obtained by all 3 groups are similar. It was only when analysing a “high molecular weight DISC” (obtained by sucrose density gradient centrifugation) that Dickens *et al.* obtained the 1:9 ratio; moreover, this was only reported for one model (BJAB).

Page 6, line 133: The authors state that the ratio for caspase-8:c-FLIP is about 7:1. This is true for the HCT116 cell line. However, in A549 and DU145 cells, the ratio is much smaller. Please clarify. This has been clarified; see page 5, lines 133-134. Overall, we see lower casp-8:FLIP ratios than in other published studies (Dickens *et al.* and Schleich *et al.* 2012). This difference may reflect the different cell line models used (heme versus epithelial), the sensitivity of mass spec for detection of FLIP and the extent of receptor stimulation used (for example, Schleich *et al.* activated 100% of Fas /CD95 receptors in their DISC IP studies, which will push the casp-8:FLIP ratio higher as the less abundant FLIP becomes depleted).

Page 6, line 135: The authors conclude that a 1:1 ratio between caspase-8 and c-FLIP is required to inhibit death receptor-mediated apoptosis. However, the c-FLIP levels are much lower than caspase-8 in many cell types (e.g. this study; Dickens *et al.*, 2012; Scaffidi *et al.*, 1999; Schleich *et al.*, 2012). How do the authors envisage that c-FLIP can block cell death at all? After all, c-FLIP can inhibit death receptor mediated apoptosis in cancer cells even when it is lower expressed than in normal tissue (Ewald *et al.*, 2011). This is an important point. First, to clarify: we conclude that a 1:1 ratio between FLIP(L) and caspase-8 inhibits apoptosis, but that a 1:3 ratio FLIP(S):caspase-8 is theoretically sufficient (although in most cell line models, FLIP(L) is the more abundant splice form). Nonetheless, FLIP levels are typically much lower as the Reviewer states – by our own unpublished quantification, ~20- and ~50-fold lower in TRAIL-sensitive and -resistant models respectively. As we previously published [Majkut *et al.* Nat Comms 2014], we believe that FLIP’s ability to compete with procaspase-8 for DISC binding despite its lower expression is at least partly explained by the higher affinity of FLIP’s death effector domain-2 (DED2) for the a1/a4 binding pocket in FADD’s DED. An additional contribution to FLIP(L)’s ability to efficiently compete with caspase-8 is that the binding between FLIP(L)’s pseudo-catalytic domain and the catalytic domain of caspase-8 is stronger than between two caspase-8 catalytic domains [Boatright *et al.*, Biochem J., 2002].

Figure 2: The calculation of the caspase-8:c-FLIP ratio is missing in the siRNA experiments. Please provide these calculations. Now presented as Fig. 2C. Second, I am not convinced that the band marked with an asterisk on panel E is ubiquitinated p43-c-FLIP_{long}. The double band could also be endogenous and Flag-tagged c-FLIP_{long} (full length). Please provide data for ubiquitination or rephrase. The exogenous FLIP in this model has no Flag-tag; however, we accept that it is speculation that this is a mono-ubiquitinated form of p43-FLIP and have removed this speculation from the figure legend. Third, the phrase “...followed by a 90 minute DISC IP.” (panel E) is misleading. Was the IP performed for 90 minutes or were the cells stimulated for 90 minutes with anti-TRAIL-R2 antibody followed by a DISC IP? Please rephrase accordingly. We have rephrased this in the figure legend (cells were stimulated for 90’).

Figure 5, panel B: What does “FLIP KO/FLIP WT” mean (labeling y-axis)? If this is the simple ratio, then the activity of caspase-8 in c-FLIP knockout cells does not reach the activity level of wildtype cells; not even after 10 hours of stimulation. If so, how are higher caspase-3/7 activity levels reached at later time points? Please explain! Apologies for any confusion here. We have re-labelled the y-axis for clarity. This graph shows the slower kinetics of caspase-8 and -3/7 activity in cells lacking FLIP following stimulation with anti-TRAIL-R2 antibody conjugated beads. This is further shown in panel C(i-ii) in cells treated with IZ-TRAIL. By later timepoints, the FLIP-deficient cells catch up as shown in panels D(i-ii); this is consistent with the ratios in panel B approaching 1 at the later timepoints.

Page 16, line 341: The authors cite a paper stating that c-FLIP does not form DED filaments. However, looking at the data in the cited paper, it is obvious that c-FLIP overexpression results in aggregates that are visible as GFP speckles (though they do not form as extensive filaments as caspase-8 DEDs). This is consistent with another study showing that isolated DEDs of murine c-

FLIP form filaments and that murine c-FLIP enhances FADD filaments in contrast to viral FLIP, which actually inhibits FADD DED filament formation (Ueffing et al., 2008). Hence, a putative inefficient oligomerization of c-FLIP might not be the explanation for reduced DISC recruitment in the absence of caspase-8. The authors might want to revisit the discussion. **We have modified this discussion point. Page 10, Lines 307-310.**

Page 16, line 359/360: The authors state that endogenous c-FLIP is detectable in the 2 MDa fraction. However, this is hardly visible. Please improve the data. **A longer exposure from a replicate Western of these samples is now presented, which more clearly shows the endogenous FLIP(S) and p43-FLIP(L).**

Page 16, line 362: "...our findings resolve the enigmatic role of FLIP(L) at the DISC." While the authors provide strong data on the dual role of c-FLIP in cells, explanations for pro cell death functions of c-FLIP have been published before (e.g. Boatright et al., 2004; Fricker et al., 2010). Therefore, the statement should be toned down. **We agree and have modified the text accordingly.**

Minor points

The authors give the concentration of the anti-TRAIL-R2 antibody as 1x, 2x and so on. Real concentrations should be given to allow for reproducibility by other investigators. **The methodology for this is described in detail in our previous publication [Majkut et al. Nat Comms, 2014]. We have described this more fully in the methods of the current paper (page 12, line 376-380).**

Page 6, line 134: Figure 1 should read Figure 1E. **Amended.**

Figure 4: Please include a description of the DED structure model including the H9 and F25 residues in the figure legend (panel D). **Amended.**

Page 12, line 276: "long-term viability assays" I would reserve this term for clonogenic assays (or similar) that run for 1-3 weeks. Here, cells were analyzed after 72 hours. Please rephrase. **Amended.**

Figure 5, panel Diii: What does "% apoptosis" mean? Which assay has been performed? Is it Annexin V/PI as in Figure 5Ciii? Please clarify. **Amended.**

Page 14, line 291: I suggest to write "... each activated receptor trimer." To make it clearer for non-specialist readers. **Amended.**

Page 14, line 310/311: Please define "these conditions". After all, the manuscript discusses that c-FLIP can have pro- as well as anti-apoptotic functions. Again, this will help non-specialist readers to follow the story. **Clarified.**

Referee #2:

Specific comments

Introduction: Prior data pertaining to the impact of FLIP on caspase-8 activity is well described. I feel it would be useful to also tell/remind readers about FLIP's ability to modulate caspase-8 specificity, as described by Pop et al, 2011 (PMID: 21235526). **This has been added.**

Fig 1 A: Were the images showing caspase-8 p24/26 from the same blots as those showing pro-casp-8 and the p41/43 species? **Yes.** If so, why are they shown in separate boxes, cropping out the region between 26 and 41kDa? **There is a non-specific band in that region that comes up with the LiCOR rabbit secondary antibody (see uncropped DISC IPs; Supplementary Information).** The positions of molecular weight markers should be shown on all blots. This will help readers consider whether, for example, the apparent lack of a FLIP(L) band in the DISC blot really reflects the absence of this protein in the DISC, or just cropping of the image below the band. **MW for all Westerns are now included.**

In Figure 2, the concentration of the scrambled siRNA should be stated in the legend or labelled in the figure. **Amended.** Why did this treatment boost caspase activity, alter the TRAIL-R2 band pattern (splicing?), and promote FLIP cleavage? **There appears some confusion here. The blot**

presented was the unbound soluble fraction from the DISC IP in panel A. Therefore, apart from the untreated (UT) sample, all other samples (including the scrambled siRNA) are from TRAIL-R2 stimulated cells. This means DR5 and DISC-bound proteins will be depleted in the unbound fraction and caspase-8 activity will increase in this fraction as it is released from the DISC (at least until caspase-8 expression falls to the 1:1 ratio with FLIP). In the revised MS, we have moved this panel to the Supplementary Figures (4E) and replaced it with the more informative CASP8:FLIP ratio requested by Reviewer 1.

Fig 2D,G, 5C: How confident are the authors that IETDase activity reflects caspase-8 activity? Caspase-3 cleaves this sequence more efficiently than caspase-8 (PMID: 17975551). This is a good point. In the revised MS, we have changed the labelling in all the figures to "IETDase activity". However, as shown in Supplementary Figure 4F, we see close correlations between IETDase activity and levels of the p41/43- and p24/26-caspase-8 cleavage products detected at the DISC, suggesting that this measurement is a good reflection of caspase-8 activity.

Fig 3A, B/Supp Fig 6, text lines 170-174: Why is less caspase-10 recruited to the DISC when caspase-8 is absent, despite the cells expressing normal levels? The authors note this but do not comment on its significance. Does this imply the existence of caspase-8/10 mixed dimers? This has previously been described for the Fas/CD95 DISC by Horn *et al.*, Cell Reports 2017. We now more clearly reference this publication in the text, which reported that procaspase-10 is recruited to the Fas DISC in a caspase-8-dependent manner and inhibits caspase-8-mediated apoptosis. Horn *et al.* suggest that caspase-10 can form mixed dimers with caspase-8. Our results are consistent with this, or that procaspase-8's scaffolding function is needed for the recruitment of procaspase-10 homodimers.

Line 161. I feel the subtitle "FLIP can interact directly with FADD in a caspase-8-independent manner" is a bit misleading. To my eyes, the main finding from Fig 3 is that FLIP recruitment to the DISC is dramatically reduced when caspase-8 (+/- 10) are absent. Yes, a tiny amount is still recruited in cells lacking these caspases, so I guess FLIP CAN be recruited in a caspase-8-independent manner, but that subtitle may give the erroneous impression that recruitment of FLIP is unaffected by caspase-8 removal. **Agreed. We have changed the subsection title to reflect this.**

Lines 178-180, re Fig 3B: "low amounts of FLIP(L) detected in the absence of procaspase-8 are cleaved by procaspase-10 and that these low levels of FLIP can be recruited by direct binding to FADD". Why would this recruitment of FLIP to the DISC in caspase-8-null cells have to be via direct interaction with FADD? Couldn't recruitment of those few FLIP molecules be via binding to the few caspase-10 molecules at the DISC, which are (as the authors note) presumably responsible for their cleavage? **Agreed. We have amended the text accordingly.**

Fig 3A/B versus E/F: Why does the presence of caspase-8 make a massive difference to FLIP recruitment to the DISC, but has only marginal impact on the ability of FLIP and FADD to interact in the nanoluciferase system? Is this because the nanoluciferase assay reports on interactions of soluble cytosolic proteins, but the DISC IPs only analyze binding events at ligated TRAIL-R2? **Yes. While the Nanoluciferase assay shows that FLIP and FADD can interact in cells, the DISC IP shows that despite this, FLIP's DISC recruitment is highly caspase-8-dependent.** Does this provide some clues about possible conformational differences between DISC-associated FADD/FLIP and the soluble cytosolic versions of those proteins? Alternatively, could the fusion of the luciferase domains to the proteins affect their folding? **The Nanoluciferase domains are connected to the death effector domains by a flexible linker specifically designed to avoid this potentially confounding effect.** Moreover, the FADD alpha-1 peptide experiments (Figure 4D) show that the orientation of the FADD-FLIP interaction in the Nanoluciferase assay is the same as that which we previously reported at the DISC [Majkut *et al.*, Nat Comms, 214], *i.e.* FADD primarily interacts with FLIP using the a1/a4 region of its DED. I think readers would appreciate the authors' explanations for the differential requirement for caspase-8, for FLIP/FADD binding in these different assays. **We have now covered this more completely on page 9, lines 267-269 and page 10, lines 305-310.**

Line 237: "neither F25A nor H9G the mutant FADD proteins...". Should "the" be removed? **Yes.**

Fig 4D/E, lines 234-244: Have the structures of the F25 and/or H9 FADD mutants been determined? Has it been confirmed that these mutations do not affect the folding of the DD? **These residues are on the surface of the FADD protein, and our molecular modelling predicts no disruption to the**

folding of FADD arising from their mutation. In pull-down and IP experiments, we have previously shown that FADD F25A preferentially interacts with FLIP's DED2 and FADD H9G preferentially interacts with Caspase-8's DED1 [Majkut et al., Nat Comms, 2014]. These results indicate that the structure of the FADD DED is maintained in these mutant proteins as, if the structures were grossly affected, they would completely lose their ability to bind to other DED proteins.

Legend to Figure 4D: The structure should be described (and referenced). This has been added.

Line 310. I didn't notice any data regarding RIPK1 processing, so presumably this statement was based on prior demonstrations of RIPK1 cleavage by caspase-8-FLIP dimers. If so, the relevant papers should be cited. References have been added.

Lines 309-311, Fig 6A "procaspase-8/FLIP(L) heterodimer has catalytic activity and can not only cleave adjacent homo- and hetero-dimers, but also necroptosis-inducing RIPK1. Under these conditions, FLIP(L) is therefore potentially both anti-apoptotic...". Why is the catalytic activity of the caspase-8/FLIP heterodimer characterized as anti-apoptotic? We have clarified this section to emphasise that this is when FLIP(L) and caspase-8 are equi-stoichiometric and there is a predominance of FLIP(L)/caspase-8 heterodimers. I understand that heterodimers are less pro-apoptotic than caspase-8 homodimers, but they're still pro-apoptotic, aren't they? According to Pop et al, the heterodimer processes procaspase-3 around half as efficiently as a caspase-8/caspase-8 homodimer. Fig 2 F/G showed that cells whose DISCs contain slightly more FLIP than caspase-8 still contain about a third as much caspase activity as cells with a higher ratio of caspase-8 to FLIP. Pop et al show that the heterodimer *can* process procaspase-3 in cell-free systems; however, because it remains tethered to the DISC in cells, the heterodimer can only cleave local substrates (e.g. other heterodimers, caspase-8 homodimers and RIPK1). The lack of caspase-3 activity under equi-stoichiometric conditions (see 30nM concentration of siCASP8 in Figures 2A-D and 10nM concentration in Supp Figure 5) suggests that this potential substrate is not proximal to the DISC and so is not cleaved by the DISC-bound heterodimer in cells.

2nd Editorial Decision

17 December 2019

Thank you for the submission of your revised manuscript to EMBO reports. We have now received the full set of referee reports that is copied below.

As you will see, all referees are very positive about the study and request only minor changes to clarify the figures.

Browsing through the manuscript myself, I noticed a few editorial things that we need before we can proceed with the official acceptance of your study.

- Thank you for supplying source data. These will be published with the manuscript, which will also address the concern from referee 2. Please note that we need one file per figure, i.e., please combine the data for Fig. 2A and 2E and for Fig. 4A and 4E into one file. The source data file currently labeled 'EV5A' in fact shows data from Fig. EV4C. Please correct.

- Your article has currently 6 figures and needs to be published as 'Article'. In this case we need a separate Results and Discussion section. Alternatively, you could of course reduce the number of main figures to 5 and move some data to an Appendix.

- You list an author with the initials 'GE-F' in the Author Contributions section but there appears to be no author with these initials. Could you please double-check this?

- Please note that per our editorial policies all data that are described need to be shown in the manuscript. Therefore, please remove the reference to "data not shown" and page 8, line 216 and show the respective data in the paper.

- Please provide all figures as individual production quality figure files (either .eps, .tif, or .jpg and one file per figure).

- Please shorten the title to max. 100 characters incl. spaces.
- Legend Figure EV3C: the legend state that the panel represents the quantification of panel C. Which panel C does this refer to? One of the main figures? Please double-check.
- Author checklist: please note that it will be published alongside the reviewer comments in the "review process file". You indicated 'NA' for most questions on statistics. Please double-check if this indeed applies.
- I attach here the synopsis image in its final size (550 pixels width). At least to my eyes, some of the text appears rather small and I suggest increasing text size and image resolution.
- I made some changes to the summary text. Could you please review the attached file and let me know whether you agree with these changes?

REFEREE REPORTS

Referee #1:

The authors have adequately addressed all my previous concerns. I congratulate the authors to this thorough study.

Referee #2:

I didn't see the Supplementary data (Appendix) file containing the full western blots amongst the files available for review. Perhaps the editor can ensure it is included if/when the manuscript is published. Also, I would have preferred the positions of MW markers to be more precisely shown with horizontal lines in addition to the numbers. But the authors addressed my other points, and those of the other reviewer (which I feel were also fair and constructive, even though they didn't occur to me on my first reading), so I don't feel these minor quibbles should prevent publication of the revised manuscript. I congratulate the authors on a carefully-performed and thorough study.

2nd Revision - authors' response

20 December 2019

The authors performed all minor editorial changes.

Corresponding Author Name: Dan Longley

Manuscript Number: EMBOR-2019-49254